# Archimedes Meets Privacy:
# On Privately Estimating Quantiles in High Dimensions Under Minimal Assumptions

**Omri Ben-Eliezer**
Department of Mathematics
MIT
omrib@mit.edu

**Dan Mikulincer**
Department of Mathematics
MIT
danmiku@mit.edu

**Ilias Zadik**
Department of Mathematics
MIT
izadik@mit.edu

## Abstract

The last few years have seen a surge of work on high dimensional statistics under privacy constraints, mostly following two main lines of work: the "worst case" line, which does not make any distributional assumptions on the input data; and the "strong assumptions" line, which assumes that the data is generated from specific families, e.g., subgaussian distributions. In this work we take a middle ground, obtaining new differentially private algorithms with polynomial sample complexity for estimating quantiles in high-dimensions, as well as estimating and sampling points of high Tukey depth, all working under very mild distributional assumptions.

From the technical perspective, our work relies upon fundamental robustness results in the convex geometry literature, demonstrating how such results can be used in a private context. Our main object of interest is the (convex) floating body (FB), a notion going back to Archimedes, which is a robust and well studied high-dimensional analogue of the interquantile range. We show how one can privately, and with polynomially many samples, (a) output an approximate interior point of the FB – e.g., "a typical user" in a high-dimensional database – by leveraging the robustness of the Steiner point of the FB; and at the expense of polynomially many more samples, (b) produce an approximate uniform sample from the FB, by constructing a private noisy projection oracle.

## 1 Introduction

Computing statistics of large, complex high-dimensional datasets under privacy constraints is a fundamental challenge in modern data science. In this work, we study *the sample complexity* of several different tasks related to estimating the quantiles of a $d$-dimensional distribution $\mathcal{D}$, from having sample access to it, under privacy constraints. Our first focus is on estimating quantiles of fixed marginals. For a random vector $X \in \mathbb{R}^d$, the quantiles of $X$ along direction $\theta$ are the quantiles of the (one-dimensional) marginal $\langle X, \theta \rangle$, which we denote by $Q_q(\langle X, \theta \rangle)$ for $q \in [0, 1]$. Our second focus is on estimating a convex region called the *floating body* of a $d$-dimensional distribution, an analogue of the 1-dimensional interquantile range exhibiting rich mathematical properties, investigated over decades of research in convex geometry and sharing connections with the work of Archimedes (see, e.g., the survey [NSW19]). Formally, the $q$-floating body of a random vector $X \in \mathbb{R}^d$ is given by

$$F_q(X) = \bigcap_{\theta \in \mathbb{S}^{d-1}} \left\{ x \in \mathbb{R}^d : \langle x, \theta \rangle \leq Q_q(\langle X, \theta \rangle) \right\}, \tag{1}$$

and in statistical language corresponds to the points of so-called *Tukey-depth* at least $1 - q$ [NSW19]. When clear from context, we also denote by $F_q(\mathcal{D})$ the floating body of the random vector $X \sim \mathcal{D}$. See Figure 1 for a simple demonstration of the floating body.

36th Conference on Neural Information Processing Systems (NeurIPS 2022).

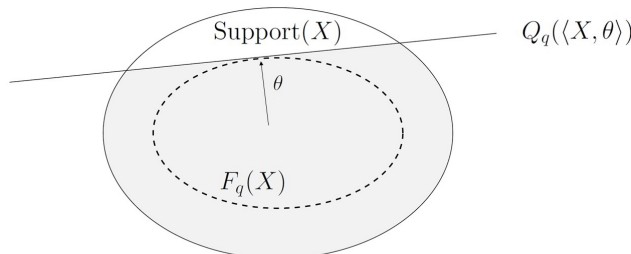

Figure 1: The floating body of a dataset $X$.

Similar to their $1$-dimensional analogues, the quantiles in high-dimensions are preferred to other descriptive statistics and location estimators due to their robustness properties [Hub81, MR97] and high breakdown points [DG07, Alo03, TP16]. As opposed to other statistics, marginal quantiles can be estimated under the mere minimal distributional assumption of having non-trivial mass around the location of the quantiles (see e.g. [AMB19, Lemma 3]). Moreover, the floating body can also be estimated in the Hausdorff distance for log-concave measures and stable laws [AR20].

Besides statistical guarantees, an essential property of an estimator in modern data science is to be private. For instance, many large healthcare datasets contain sensitive patient information [DE13, ZLZ+17, FLJ+14, YKÖ17] and it is crucial for an estimator to balance the trade-off between protecting the privacy of the input while producing accurate statistical results.

Differential privacy (DP) is the leading notion in quantifying privacy guarantees of a randomized algorithm over an input dataset. In this work, we focus on algorithms satisfying *pure* DP guarantees. Given a parameter $\varepsilon > 0$, $\varepsilon$-differential privacy guarantees that the probability of a given output cannot change more than a multiplicative factor $e^\varepsilon$ after the arbitrary change of the input data of one data item (e.g., a single user record in a large database). More generally, $\varepsilon$-DP can be defined as follows with respect to the Hamming distance $d_H(X, X') := |\{i \in [n] | X_i \neq X_i'\}|$, defined for two $n$-tuples of samples $X = (X_1, \ldots, X_n), X' = (X_1', \ldots, X_n') \in (\mathbb{R}^d)^n$.

**Definition 1** ($\varepsilon$-DP). A randomized algorithm $\mathcal{A}$ is $\varepsilon$-differential private if for all subsets $S \in \mathcal{F}$ of the output space $(\Omega, \mathcal{F})$ and $n$-tuples of samples $X_1, X_2 \in (\mathbb{R}^d)^n$, it holds that

$$\mathbb{P}\left(\mathcal{A}(X_1) \in S\right) \leq e^{\varepsilon d_H(X_1, X_2)} \mathbb{P}\left(\mathcal{A}(X_2) \in S\right). \tag{2}$$

While there has been a lot of work on differential privacy, it usually analyzes algorithms under worst-case assumptions. The most closely related to our work is the line of research on privately outputting an interior point of the convex hull of $n$ input points in high dimensions for a worst-case input [BMNS19, KMST20, SS21, KSS20]. Only recently there has been a lot of statistical work analyzing differentially private estimators under distributional assumptions for various statistical tasks [BCS15, DJW18, BCSZ18b, KSSU19, KLSU19, SU19, ZKKW20, CKS20, TVGZ20, KSU20]. On the topic of the present work, [TVGZ20] established the exact trade-off between privacy and accuracy of median estimation in one dimension. One of the main motivations of the present work is to make a first step towards understanding the high-dimensional counterparts of these privacy-accuracy tradeoffs, by proposing the floating body as a natural estimate under privacy constraints in high dimensions. In particular, the convexity of the floating body allows us to invoke powerful robustness theorems, originating from decades of research in convex geometry. It is this combination, of classical theorems with the modern framework of DP, which yields several new and non-trivial results.

## 1.1 Our Contributions

**The minimal assumptions.** It is known that even in one dimension, private quantile estimation is impossible for arbitrary measures [TVGZ20]. Thus, some assumptions are required. In this work we make the following minimal assumptions on the underlying distribution.

**Definition 2.** We say that a distribution $\mathcal{D}$ supported on $\mathbb{R}^d$ has a $q$-admissible law with parameters $R_{\max}, R_{\min}, r, L > 0$, if, for every $\theta \in \mathbb{S}^{d-1}$, the following four conditions hold:

    1. $Q_q(\langle X, \theta \rangle) \in [-R_{\max}, R_{\max}]$.

2. For some $c \in \mathbb{R}^d$, it holds that $|Q_q(\langle X, \theta \rangle) - \langle c, \theta \rangle| \geq R_{\min}$.

3. The density $f_\theta$ of $\langle X, \theta \rangle$ exists and satisfies $f_\theta(t) \geq L$ whenever $t \in [Q_q(\langle X, \theta \rangle) - r, Q_q(\langle X, \theta \rangle) + r]$.

4. If $\{X_i\}_{i=1}^n$ are *i.i.d.* as $\mathcal{D}$, then with high probability $\|X_i\|_2 \leq \mathrm{poly}(n, d)$, for all $i \in [n]$.

The set of all such distributions is denoted by $A_q(R_{\max}, R_{\min}, r, L)$.

We now expand further on the claimed "minimality" of the above assumptions. First, note that assumptions (1) and (3) are actually necessary for consistent private quantile estimation. This is an outcome of the tight lower bound for the one-dimensional case (see the main result of [TVGZ20]), where it is established that the minimal sample complexity to privately learn one quantile of a distribution explodes to infinity whenever there is no bound on the possible quantile values ($R = R_{\max} = \infty$) or the distribution has no mass around the quantile ($L = 0$). We highlight that the latter property (of having a positive density around the quantile) is a requirement for optimal estimation rates (and a standard assumption) for even *non-private* one-dimensional quantile estimation (e.g., see [VdV00, Chs. 21, 25.3]). Now, assumption (2) is equivalent to the assumption that the floating body contains a ball of radius $R_{\min}$. Again, if $R_{\min} = 0$, the floating body has an empty interior, and the floating body's estimation becomes a degenerate task. Finally, assumption (4) is a very mild boundedness technical condition that our data is arbitrarily polynomially-bounded, which almost does not hurt generality. For example, for the "heavy-tailed" Cauchy random vectors this assumption holds for the polynomial $dn$.

**Private many-quantiles estimation.** Let us introduce a useful convention: we identify a fixed vector $X = (X_1, \ldots, X_n) \in (\mathbb{R}^d)^n$ with a random vector in $\mathbb{R}^d$, chosen uniformly from $(X_1, \ldots, X_n)$. In other words we identify a "sample" $X$ with the empirical distribution over its $d$-dimensional coordinates. Now, in [AR20] it was established that for any symmetric log-concave measure $\mathcal{D}$, if one has $n$ i.i.d. samples $X = (X_1, \ldots, X_n) \in (\mathbb{R}^d)^n$ from $\mathcal{D}$, then the empirical measure on the samples satisfies for $Y \sim \mathcal{D}$ with probability at least 0.9,

$$\delta_q(X, Y) := \sup_{\theta \in \mathbb{S}^{d-1}} |Q_q(\langle X, \theta \rangle) - Q_q(\langle Y, \theta \rangle)| \leq \alpha, \tag{3}$$

for some $n = \tilde{O}(d/\alpha^2)$. Here, and throughout the paper, $\tilde{O}$ means we omit logarithmic factors, in all parameters. One can easily generalize the above result for any $q$-admissible law (as described in Definition 2) (see Supplementary material).

We now proceed with an observation. For any integer $M > 0$, and for any subset of $M$ directions, say that one wants to estimate the $q$-quantiles of the distribution along each of the $M$ directions. A simple "low-dimensional" approach would be as follows; one can take $n$ samples and then take the empirical $q$-quantiles of the projection of the samples along each of the $M$ directions. Using standard concentration results having $\tilde{O}(1/\alpha^2)$ fresh samples per direction suffices for estimation in the corresponding direction, leading to the desired estimation guarantee with $n = \tilde{O}(M/\alpha^2)$ samples. We note that for each specific direction $\tilde{O}(1/\alpha^2)$ samples are known to be necessary [TVGZ20, Proposition 3.7]. Comparing with the guarantee of [AR20] in (3) we are lead to an interesting high-dimensional statistical phenomenon: without privacy constraints, *simultaneously* estimating $M > d$ marginal quantiles in high-dimensions requires less samples than the natural (but perhaps naive) approach of projecting the points and then calculating the empirical quantile on each direction.

Our first contribution is to reveal the private analogue of the above high-dimensional phenomenon. Consider the task now of *private estimation of $M$ quantiles*, i.e. of $Q_q(\langle Y, \theta_i \rangle), Y \sim \mathcal{D}$ for $M$ different directions $(\theta_i)_{i=1}^M$. The "naive" approach would be, as before, to first project along the different directions, but then, instead of calculating the (non-private) empirical quantiles of the projected points, one would apply the optimal private algorithm of [TVGZ20] to each projection, separately.[1] This would lead to the guarantee in (3) with sample complexity (assuming, for simplicity, *here and throughout the contribution section* all parameters besides $M, \varepsilon, \alpha$ are constant)

$$n = O\left(\frac{M}{\alpha^2} + \frac{M}{\varepsilon \alpha}\right). \tag{4}$$

---

[1] The algorithm from [TVGZ20] is technically only designed for median estimation, but with the natural modifications the same algorithm and analysis can be straightforwardly extended for $q$-quantile estimation.

Our first main result is an improved private rate for the estimation of $M$ quantiles which appropriately privatizes the high-dimensional result of [AR20].

**Theorem 3** (informal). *There exists an $\varepsilon$-differentially private algorithm that draws*

$$n = \tilde{O}\left(\frac{d}{\alpha^2} + \frac{M}{\varepsilon\alpha}\right)$$

*samples from any admissible distribution $Y \sim \mathcal{D}$ and produces an estimate $\hat{m}$ that satisfies $\max_{i=1,\ldots,M} |Q_q(\langle Y, \theta_i \rangle) - \hat{m}_i| \leq \alpha$ with probability at least $0.9$.*

Notice that the rate described in Theorem 3 is always at least as good as the rate in (4), and that it is strictly better when $M > d$ and $\varepsilon > \alpha$. This constitutes indeed a private analogue of the high-dimensional phenomenon mentioned above; privately estimating $M > d$ can require much less samples than simply projecting and applying the one-dimensional machinery.

**Private interior point.**   Next, we attempt to privately produce point estimates of the floating body $F_q(Y)$ itself. Our first task is to output an interior point of $F_q(Y)$, a relatively easy task without privacy concerns [AR20], with a clear statistical motivation of outputting a "typical datapoint" in high-dimensions. We show that with polynomial in $d$ samples one can achieve such a guarantee with small error. There has been closely related (but not fully comparable) work on this task in the worst case regime, which we discuss in depth in Section 1.2.

For our result we leverage the Steiner point of the floating body, denoted $S(F_q(Y))$ (see (7) for the definition). The Steiner point has been widely studied in the context of *Lipschitz selection* [BL00]. In the problem of Lipschitz selection, one looks to consistently select a point from the interior of each convex body, in a way which is robust to perturbations of the body. The Steiner point is celebrated for being the optimal Lipschitz selector in this setting. Just recently, this construction proved to be instrumental in the resolution of the *chasing nested convex sets* problem [BKL$^+$20, Sel20, AGTG21], a problem in online learning, related to Lipschitz selection. It is precisely this optimal robustness that makes the Steiner point insensitive to individual sample points, a desirable property in the design of DP algorithms. To the best of our knowledge, this is the first time the Steiner point and its properties have been used it to build differentially private estimators, and we believe it might be useful in other similar problems.

**Theorem 4** (informal). *There exists an $\varepsilon$-differentially private algorithm that draws*

$$n = \tilde{O}\left(\frac{d}{\alpha^2} + \frac{d^{2.5}}{\varepsilon\alpha}\right)$$

*samples from any admissible distribution $Y \sim \mathcal{D}$ and produces an estimate $\hat{m} \in \mathbb{R}^d$ that satisfies $\|\hat{m} - S(F_q(Y))\|_2 \leq \alpha$ with probability at least $0.9$.*

In comparison, the best known dependence of $n$ on the dimension in the worst case literature is of order $\tilde{O}(d^4)$ for pure $\varepsilon$-DP [KSS20] and of order $\tilde{O}(d^{2.5})$ for the weaker notion of approximate $(\varepsilon, \delta)$-DP [BMNS19].[2] Thus, Theorem 4 achieves essentially the same dependence as the best known approximate DP result while enjoying the stronger guarantees of pure DP. We note again that the results are not fully comparable; see Section 1.2 for a thorough discussion.

**Private sampling from the floating body.**   Outputting an, a-priori, arbitrary point from the floating body comes with possible drawbacks, for example biasing the output point towards or away from its boundary. This can lead to inaccurate statistical conclusions if a more detailed description of the "typical datapoints" is needed. For this reason, we undertake the harder task of outputting a *uniform sample*. The *sampling from convex bodies* problem has a long and rich history with many deep results, see [KLS95, KLS97, LV06] for some prominent examples. In many cases, the computation preformed by the sampling algorithm reduces to iteratively projecting points into the convex body, along the trajectory of an appropriate Markov chain [BEL18, Leh21].

It is precisely the idea described above which serves as the working engine behind our private sampling algorithm. To cope with privacy constraints, we capitalize on known robustness properties

---

[2]See Supplementary material for a definition and comparison between the different privacy notions.

of the projection operator (see (8)) on convex bodies, as proved in [AW93]. Remarkably, when the projected point is fixed and one considers the convex body as the variable, the projection operator is known to be 1/2-Hölder with respect to the Hausdorff distance. This allows us to build a noisy projection oracle for the floating body, in a private fashion. Combining the noisy oracle with known results leads to a new sampling algorithm, which operates under pure differential privacy, and with guarantees in the quadratic Wasserstein distance, $W_2$ (see Section D.2 for the definition).

**Theorem 5** (informal). *Let $\mu_{F_q(Y)}$ be the uniform measure on the floating body. There exists an $\varepsilon$-DP algorithm that draws*

$$n = \tilde{O}\left(\frac{d^2}{\alpha^{14}} + \frac{d^4}{\varepsilon^2 \alpha^8}\right)$$

*samples from any admissible distribution $Y \sim \mathcal{D}$, and whose output, $\hat{\mu}_{F_q(Y)}$, satisfies*

$$\frac{1}{d} W_2^2(\hat{\mu}_{F_q(Y)}, \mu_{F_q(Y)}) \leq \alpha.$$

We note that the $\frac{1}{d}$ normalizing factor in Theorem 5 serves as the correct scale for the $W_2$ metric which typically scales as square-root of the dimension, see [Leh21] for further discussion on this.

## 1.2 Further Comparison with [BMNS19, KSS20]

The most closely related work to our point estimators of the floating body (Theorems 4 and 5) are by Beimel et al. [BMNS19] and Kaplan et al. [KSS20]. Both of these works address under worst-case input the private *interior point* problem: namely, given a (worst case) set of $n$ points $x_1, \ldots, x_n$ in $d$ dimensions, the task is to privately output a point that lies within the convex hull of these points. While the non-private analogue is trivial (since one can always output, say, $x_1$), the problem becomes interestingly non-trivial when privacy considerations are introduced. To achieve such guarantees, the set of points is assumed to be a subset of a finite grid of possible points $U^d$ (where $U$ is some finite universe). This is a necessary condition for the existence of successful private estimators under worst-case assumptions [BNS13, BNSV15]), whereas this assumption is not required in our case. On the other hand, due to the use of the extension lemma, our algorithms do not have any explicit bound on the running time (see Section 1.3 below), unlike the situation in the worst case works, whose running time is generally exponential in $d$.

The first work [BMNS19] solves the private center point problem under approximate $(\varepsilon, \delta)$-differential privacy, a weaker privacy guarantee than $\varepsilon$-differential privacy, assuming that the dataset is of size at least some $n = \tilde{O}(d^{2.5} \log^*(|U|))$ (in the $\tilde{O}$ term here we suppress dependencies in $\varepsilon, \delta$, and lower order terms). The second work [KSS20] solves the problem under pure differential privacy (as in our paper) for $n = \tilde{O}(d^4 \log |U|)$, and obtains an improved $O(n^d)$ running time using $O(d^4 \log |U|)$ samples under approximate DP. The $n = \tilde{O}(d^4 \log |U|)$ bound is currently the best known sample complexity under pure DP for this worst-case task. In both [BMNS19] and [KSS20], the authors' approach is to output a point of high Tukey-depth with respect to the set of the $n$ points, which can be interpreted as outputting an element of the $q$-floating body of the $n$ points for some appropriate, and perhaps data-dependent, value of $q$.

While our considered settings are similar but incomparable, we recall that our Theorem 4 obtains pure $\varepsilon$-differentially private guarantees that for any $q$ outputs a point of the $q$-floating body with $\tilde{O}(d^{2.5})$ samples. Finally, we note that matching the sample complexity of the best $(\varepsilon, \delta)$-approximate private algorithm with an $\varepsilon$-private algorithm has been challenging in the privacy literature, e.g., in the context of high-dimensional Gaussian mean estimation [KLSU19, KSU20], and in fact provably impossible in the worst-case context for many problems of interest; see [BMNS19] for more details.

## 1.3 General Approach

Our approach towards building all $\varepsilon$-private algorithms in this work is based on an appropriate use of a Lipschitz extension tool called the Extension Lemma [BCSZ18a, BCSZ18b]. This lemma was also the main tool behind the construction of the private median estimator in [TVGZ20]. One of the main hurdles in constructing sample-efficient $\varepsilon$-differentially private algorithms under distributional assumptions is that the privacy constraint needs to hold for all input data-sets, while the accuracy is based on the behavior of the algorithm on "with-high-probability" or typical data-sets. The Extension

Lemma completely resolves this issue and allows the algorithm designer to focus on ensuring privacy solely on the typical data-sets[3] Then a (privacy-preserving) Lipschitz Extension argument takes care of extending the private algorithm on typical-inputs to a private algorithm on all possible inputs, while maintaining the original algorithm's output on the typical inputs (see Proposition 6 for more details). Leveraging this tool, our results are based on a two-step procedure.

1. First we consider the (non-private) estimator of interest. For example, this is the Steiner point of the floating body for Theorem 4 and the projection of a point to the floating body for Theorem 5. Then we define a *typical set* of possible input datasets $X = (X_1, \ldots, X_n) \in (\mathbb{R}^d)^n$ which is realized with high probability over all inputs drawn from admissible distributions. Following this we establish Lipschitzness properties of the non-private estimator seen as a function of the input $X$ *with respect to the Hamming distance between two inputs* of the (non-private) estimator, when defined only on inputs from the typical set. Such robustness properties can be non-trivial and to do so we use interesting tools from convex geometry to establish them for the Steiner point [BL00], and for the projection of a point [AW93]. Notably, none of these estimators is (non-trivially) Lipschitz with respect to the Hamming distance of the input, unless the input is on the typical set. Using now the Lipschitzness properties combined with classic differential privacy ideas we can "privatize" the estimator by applying the (flattened) Laplace mechanism [TVGZ20], while ensuring high accuracy.

2. With the Extension Lemma, we extend our estimator to be defined and private on all possible inputs. The Extension Lemma also guarantees that the estimator remains identical when the input is from the typical set. These allow to obtain a globally private algorithm with the same accuracy guarantee.

## 2   Getting Started

**Extension Lemma**   Let us consider an arbitrary $\varepsilon$-differentially private algorithm defined on $n$ samples in $\mathbb{R}^d$ as input and belonging in some set $\mathcal{H} \subseteq (\mathbb{R}^n)^d$. Then the Extension Lemma guarantees that the algorithm *can be always extended* to a $2\varepsilon$-differentially private algorithm defined for arbitrary input data in $(\mathbb{R}^n)^d$ with the property that if the input data belongs in $\mathcal{H}$, the distribution of output values is exactly the same with the original algorithm. We note that the result in [BCSZ18a] is "generic", in the sense of applying to any input metric space and output probability space, but here we present it for simplicity only on inputs from $\mathbb{R}^n$ and equipped with the Hamming distance, $d_H$.

**Proposition 6** (The Extension Lemma, Proposition 2.1, [BCSZ18a])**.** *Let $\hat{\mathcal{A}}$ be an $\varepsilon$-differentially private algorithm designed for input from $\mathcal{H} \subseteq (\mathbb{R}^d)^n$ with arbitrary output measure space $(\Omega, \mathcal{F})$. Then there exists a randomized algorithm $\mathcal{A}$ defined on the whole input space $(\mathbb{R}^d)^n$ with the same output space which is $2\varepsilon$-differentially private and satisfies that for every $X \in \mathcal{H}$, $\mathcal{A}(X) \stackrel{d}{=} \hat{\mathcal{A}}(X)$.*

**Quantiles and Floating Bodies**   For $X$ a random variable over $\mathbb{R}$, and $q \in (0, 1)$, the $q$-*quantile* of $X$, which we denote $Q_q(X)$, is the smallest number in $\mathbb{R}$ satisfying $\mathbb{P}(X \leq Q_q(X)) \geq q$. Our main object of interest is a high-dimensional analogue of the quantile, called the convex floating body, defined in (1). Notice, that for $q < \frac{1}{2}$, $F_q(X)$ is necessarily empty and unfortunately it may sometimes by empty for $q \geq \frac{1}{2}$. Assumption 2. in Definition 2 ensures that $F_q(\mathcal{D}) \neq \emptyset$ and moreover that $F_q(\mathcal{D})$ contains a ball of radius $R_{\min}$, centered at $c$ (see Supplementary material for more details).

**Useful Distances**   The following pseudo-metric measures the maximum difference between the $q$-quantiles of two (random) vectors.

**Definition 7** ($q$-distance)**.**   Let $q \in (\frac{1}{2}, 1)$ and let $A \subset \mathbb{S}^{d-1}$. If $X$ and $Y$ are two random vectors in $\mathbb{R}^d$, their $q$-distance, with respect to $A$ is defined by $\delta_q(X, Y; A) := \sup\limits_{\theta \in A} |Q_q(\langle X, \theta \rangle) - Q_q(\langle Y, \theta \rangle)|$.

If $A = \mathbb{S}^{d-1}$, we abbreviate $\delta_q(X, Y; A) = \delta_q(X, Y)$.

In the sequel, we will consider two types of set $A$, in Definition 7. First, finite sets of directions, which correspond to privately estimating a finite (but potentially large) number of quantiles. Second

---

[3]While this work is entirely focused on sample complexity guarantees, we note that the Extension Lemma comes in principle with no explicit termination time guarantee (see [BCSZ18a] and Appendix F).

the whole set of possible directions $A = \mathbb{S}^{d-1}$. In this case where $\delta_q(X, Y; A) = \delta_q(X, Y)$ bounds on $\delta_q(X, Y)$ are strong enough to yield allow estimation of high-dimensional statistics of $F_q(X)$, such as the Steiner point of the body or the projection operator to the body.

To further elaborate on this remark, let us recall that there is another natural convex geometric way to measure distances between convex bodies, *the Hausdorff distance*. For two convex bodies $K, K' \subset \mathbb{R}^d$, their Hausdorff distance is $\delta_{\mathrm{Haus}}(K_1, K_2) := \inf\{t > 0 | K \subset K' + tB_d \text{ and } K' \subset K + tB_d\}$, where $B_d$ is the unit Euclidean ball in $\mathbb{R}^d$. The following connection between $\delta_{\mathrm{Haus}}$ and $\delta_q$ holds.

**Lemma 8.** [Bru18, Lemma 5] *Let $q \in (\frac{1}{2}, 1)$ and let $X$ and $Y$ be two random vectors in $\mathbb{R}^d$. Suppose that $F_q(X)$ contains a ball of radius $R_{\min}$ and is contained in a ball of radius $R_{\max}$, for some $R_{\max}, R_{\min} > 0$. If $\delta_q(X, Y) \leq \frac{R_{\min}}{2}$ then, $\delta_{\mathrm{Haus}}(F_q(X), F_q(Y)) \leq \frac{3R_{\max}}{R_{\min}} \delta_q(X, Y)$.*

# 3   A Meta Algorithm for Hölder Queries

We are now ready to present our main result. In the introduction, we highlighted three results of our work, all of which follow from a meta-theorem resulting in a differentially private algorithm for querying generic Hölder statistics of the floating body with respect to the $\delta_q$ norm. Since our result is general, it requires certain notation.

**Approximate Hölder queries**   Let $p \in [1, \infty]$, $h \in (0, 1]$, and $M \in \mathbb{N}$. Denote by $\mathcal{C}_d$ the space of all convex bodies in $\mathbb{R}^d$, and assign $M$ to be the dimension of the output space, equipped with the $L_p$-norm. We say that a map $f : \mathcal{C}_d \to \mathbb{R}^M$ is $h$-Hölder (with constant $K > 0$, and with respect to a set $A \subset \mathbb{S}^{d-1}$), or simply Lipschitz when $h = 1$, if

$$\|f(F_q(X)) - f(F_q(Y))\|_p \leq K\delta_q(X, Y; A)^h, \tag{5}$$

where $X$ and $Y$ are random vectors. One example to keep in mind is when $X$ and $Y$ are both the empirical distributions of two samples that satisfy $d_H(X, Y) = 1$. In this case we establish that, when drawn from admissible distributions, with high-probability $\delta_q(X, Y; A)$ is small (see Lemma 21). Thus, (5) will imply a low sensitivity condition for $f$, a desirable property for the design of differentially private algorithms to approximate $f$.

It turns out that many of the queries we study, like the Steiner point, are Hölder with respect to the Hausdorff distance, not the $\delta_q$ metric. In light of Lemma 8, it will sometimes be convenient to restrict the domain in which (5) holds. In particular, for a fixed admissible class of measures, $A_q(R_{\max}, R_{\min}, r, L)$, we shall enforce the condition that the floating bodies contain, and are contained in, a ball, as well as require some a-priori upper bound on the $\delta_q$ distance.

**Definition 9** (Approximate Hölder functions for the class $A_q(R_{\max}, R_{\min}, r, L)$). We say that $f$ is *approximate $h$-Hölder* if for all $X, Y$, which satisfy that for some $a \in \mathbb{R}^d$, which could depend on $X$,

$$B(a, R_{\min}/2) \subseteq F_q(X) \subseteq B(0, R_{\max} + r) \text{ and } \delta_q(F_q(X), F_q(Y)) \leq \frac{R_{\min}}{4},$$

(5) holds.

Since our private algorithm will be extended from a restricted algorithm on a typical set (recall the plan from Section 1.3) using Proposition 6, there is no loss of privacy when considering approximate Hölder functions, as long as the desired conditions hold with high probability over the sample.

**The main result.**   We are now prepared to state our main theorem. All results mentioned in the preceding sections will follow by working with suitable approximate Hölder functions.

**Theorem 10.** *Fix $q \in (1/2, 1)$ and assume that $\mathcal{D} \in A_q(R_{\max}, R_{\min}, r, L)$. Further, for $h, K > 0$ and $A \subset \mathbb{S}^{d-1}$, let $f : \mathcal{C}_d \to \mathbb{R}^M$ be an approximate $h$-Hölder function with constant $K$, with respect to $A$. Then, there is an $\varepsilon$-differentially private algorithm $\mathcal{A}$ which for input $X = (X_1, \ldots, X_n)$, sampled i.i.d from $\mathcal{D}$, satisfies for all $\alpha < \min\{1, K\}\frac{\min\{r, R_{\min}\}}{2}$ that $\mathbb{P}(\|\mathcal{A}(X) - f(F_q(\mathcal{D}))\|_p \leq \alpha) \geq 1 - \beta$, for some $n$ such that*

$$n = \tilde{O}\left(\frac{K^{2/h}\left(d + \log\left(\frac{4}{\beta}\right)\right)}{\alpha^{2/h}L^2} + \frac{WK^{1/h}(\log\left(\frac{1}{\beta}\right)^{1/h}M^{1/h})}{(\varepsilon\alpha)^{1/h}L} + \frac{WM^{1/h}}{(\varepsilon\min\{r, R_{\min}\})^{1/h}L}\right),$$

*where* $W = \min\{d, \log|A|\} + \log\left(\frac{1}{\beta}\right).$

The obtained rate may seem complicated; this is to be expected, given the generality of our results and the number of parameters. In the next section we demonstrate several concrete uses of Theorem 10, which show how the rate simplifies in various interesting settings.

## 4   Applications

Here we describe the main applications of Theorem 10. As mentioned we apply the theorem using suitable approximate Hölder functions.

**Simultaneous estimation of quantiles**   Fix $q \in (0,1)$ and let $A \subset \mathbb{S}^{d-1}$, with $|A| = M$. Define the multiple $M$-query function which on a random vector $X$, equals $f_A(X) = \{Q_q(\langle X, \theta \rangle)\}_{\theta \in A}$.

It is immediately seen that $f_A$ is a Lipschitz function[4], with constant 1, in the $L_\infty$ ($p = \infty$) norm:

$$\|f_A(X) - f_A(Y)\|_\infty = \max_{\theta \in A} |Q_q(\langle X, \theta \rangle) - Q_q(\langle Y, \theta \rangle)| = \delta_q(X, Y; A). \tag{6}$$

We thus have the following result.

**Corollary 11.** *Let $\mathcal{D} \in A_q(R_{\max}, R_{\min}, r, L)$ be an admissible measure on $\mathbb{R}^d$. Then, there is an $\varepsilon$-differentially private algorithm $\mathcal{A}$ which for input $X = (X_1, \ldots, X_n)$, sampled* i.i.d *from $\mathcal{D}$, satisfies for all $\alpha < \frac{\min\{r, R_{\min}\}}{2}$, $\mathbb{P}(\|\mathcal{A}(X) - f_A(\mathcal{D})\|_\infty \leq \alpha) \geq 0.9$, for*

$$n = \tilde{O}\left( \frac{d}{\alpha^2 L^2} + \frac{M}{\varepsilon L \alpha} + \frac{M}{\varepsilon L \min\{r, R_{\min}\}} \right).$$

*Proof.* The observation in (6) shows that $f_A$ is a Lipschitz function with constant 1. The result now follows by invoking Theorem 10 for the $L_\infty$ norm, with $K, h = 1$, $\beta = 0.1$, and $W = \log(M)$.  □

**Privately returning an interior point: the Steiner point.**   Given Theorem 10 it will be enough to show that one can select a point from a convex body in a Lipschitz way. This naturally leads to the Steiner point, a widely studied object in Lipschitz selection. For a convex body $K \subset \mathbb{R}^d$, define by

$$S(K) := d \int_{\mathbb{S}^{d-1}} \theta h_K(\theta) d\sigma, \tag{7}$$

its Steiner point, where $\sigma$ is the normalized Haar measure on $\mathbb{S}^{d-1}$, and the support function of $K$, $h_K : \mathbb{S}^{d-1} \to \mathbb{R}$, is defined by $h_K(\theta) = \max_{x \in K} \langle x, \theta \rangle$.

The following result follows from well-known results in convex geometry and from Lemma 8.

**Lemma 12.** *The Steiner point $S : \mathcal{C}_d \to \mathbb{R}^d$ is an approximate Lipschitz function, with constant $6\sqrt{d}\frac{R_{\max}+r}{R_{\min}}$, which satisfies $S(K) \in K$ for every convex body $K$.*

We immediately get the following corollary to Theorem 10.

**Corollary 13.** *Let $\mathcal{D} \in A_q(R_{\max}, R_{\min}, r, L)$ be an admissible measure on $\mathbb{R}^d$. Then, there is an $\varepsilon$-differentially private algorithm $\mathcal{A}$ which for input $X = (X_1, \ldots, X_n)$, sampled* i.i.d *from $\mathcal{D}$, satisfies for all $\alpha < \frac{\min\{r, R_{\min}\}}{2}$, $\mathbb{P}(\|\mathcal{A}(X) - S(F_q(\mathcal{D}))\|_2 \leq \alpha) \geq 0.9$, for some*

$$n = \tilde{O}\left( d^2 \frac{(R_{\max}+r)^2}{R_{\min}^2 \alpha^2 L^2} + d^{2.5} \frac{R_{\max}+r}{R_{\min}} \left( \frac{1}{\varepsilon L \alpha} + \frac{1}{\varepsilon L \min\{r, R_{\min}\}} \right) \right).$$

*Proof.* Consider the Steiner point $S : \mathcal{C}_d \to \mathbb{R}^d$. Lemma 12 states that $S$ is an approximate Lipschitz function with constant $6\sqrt{d}\frac{R_{\max}+r}{R_{\min}}$, for every $X, Y \in \mathcal{H}_C$. The result follows by invoking Theorem 10 for the Euclidean norm, with $K = 6\sqrt{d}\frac{R_{\max}+r}{R_{\min}}$, $M = d$, $h = 1$, $\beta = 0.1$, and $W = \tilde{O}(d)$.  □

---

[4]Strictly speaking, $f_A$ is not a function of the floating body, but of the sample itself. However, with a trivial adaption, it still fits nicely within our framework.

**Private projection and sampling.** Let $K \subset \mathbb{R}^d$ be a convex body. Define the projection operator, $P_K : \mathbb{R}^d \to \mathbb{R}^d$ by,

$$P_K(x) = \arg\min_y \{y \in K | \|y - x\|\}. \tag{8}$$

We shall establish that, for admissible distributions $\mathcal{D}$ and for any point $x$, one can privately estimate $P_{F_q(\mathcal{D})}(x)$ with polynomially many samples. This follows by coupling a classic result in convex geometry, [AW93, Proposition 5.3], concerning the stability of projections with Lemma 8.

**Lemma 14.** *Fix $x \in \mathbb{R}^d$ and consider $P_K(x) : \mathcal{C}_d \to \mathbb{R}^d$. Then $P_K(x)$ is an approximate $\frac{1}{2}$-Hölder function with constant $5\sqrt{\frac{(\|x\|_2 + R_{\max} + r)(R_{\max} + r)}{R_{\min}}}$.*

Now, to sample from the body $K$, for $\eta > 0$, we define the following discretized Langevin process:

$$X_{t+1} = P_K(X_t + \eta g_t), \quad X_0 = S(K),$$

where $\{g_t\}_{t \geq 0}$ are *i.i.d.* standard Gaussians and $S(K)$ is the Steiner point of $K$. It is well known (see for example [Leh21, Theorem 2]) that this process mixes rapidly, in the Wasserstein distance. By applying the known results about the mixing time of the Langevin process, and by taking account of the inherent noise introduced by the privacy constraints, we prove the following result.

**Corollary 15.** *Let $\mathcal{D} \in A_q(R_{\max}, R_{\min}, r, L)$ be an admissible measure on $\mathbb{R}^d$ and let $U_q$ be a random vector which is uniform on $F_q(\mathcal{D})$. Assume that $F_q(\mathcal{D})$ contains a ball of radius $R_{\min}$ around the Steiner point $S(K)$. Then, there is an $\varepsilon$-differentially private algorithm $\mathcal{A}$ which for input $X = (X_1, \dots, X_n)$, sampled i.i.d from $\mathcal{D}$, satisfies for all $\alpha < \frac{\min\{1, r, R_{\min}\}}{2}$,*

$$\frac{1}{d} W_2^2(\mathcal{A}(X), U_q) \leq \alpha,$$

*for some*

$$n = \tilde{O}\left( \frac{\text{poly}(R_{\max} + r + 1)}{\text{poly}(R_{\min})} \left( \frac{d^2}{\alpha^{14} L^2} + \frac{d^4}{\varepsilon^2 \alpha^8 L} + \frac{d^4}{\varepsilon^2 \alpha^2 \min\{r, R_{\min}\}^2 L} \right) \right).$$

Corollary 15 requires that the floating body contains a ball, centered at the Steiner point. The reason for this assumption is that the discretized Langevin process is initiated at the Steiner point, and the distance from the initialization to the boundary of $F_q(\mathcal{D})$ will determine the mixing time of $X_t$. We chose the Steiner point as an the initial point because, by Theorem 13, we can privately approximate it. Moreover, the Steiner point tends to lie "deeply" in the interior of the convex body, although exact estimates seem to be unknown for the general case [Sch93, Section 5.4] (however see [Shv04, Theorem 1.2] for a similar construction with relevant guarantees).

To improve performance, one might impose some extra assumptions. For example, if the distribution $\mathcal{D}$ is symmetric around its mean, then the Steiner point will be at the center of $F_q(\mathcal{D})$. Another option is to assume that $F_q(\mathcal{D})$ contains a known point, like the origin, in which case we can initialize $X_0 = 0$. We chose to state Corollary 15 this way to make it as general as possible.

Such sampling schemes are related to optimization of convex functions and could be of further interest. We expand this discussion in the supplementary material.

## 5 Proof Sketch of Theorem 10

To provide intuition, we now briefly describe how one can use the two-stage procedure from Section 1.3 to obtain Theorem 10, from which all our applications follow. We fix an approximate $h$-Hölder query $f(F_q(Y)), Y \sim \mathcal{D}$ and assume for simplicity that it is Hölder w.r.t. $\delta_q(X, Y)$, that is $A = \mathbb{S}^{d-1}$.

**Obtaining a "good" (non-private) estimator** Our first step is to obtain a (non-private) estimate of the query. For that, we sample $n$ independent points $\{X_i\}_{i=1}^n \sim \mathcal{D}$, and define $X$ the uniform empirical measure over $(X_1, \dots, X_n)$, for which we compute $f(F_q(X))$. In terms of accuracy, by an appropriate generalization of the arguments in [AR20] to apply for any admissible distribution we have $\delta_q(X, Y) \leq (\alpha/K)^{1/h}$ for some $n = \tilde{O}(dK^{1/h}/\alpha^{2/h})$. Notice now that, by admissibility, and the discussion in Section A.1 it can be easily checked that $F_q(Y)$ satisfies the necessary geometric condition; it contains a ball and is contained in a ball, both of "controlled" radius. Hence, the approximate Hölderness implies $\|f(F_q(X)) - f(F_q(Y))\|_p \leq K \delta_q(X, Y)^h = \alpha$.

**Designing the private estimator on a typical set.**  Our goal now turns to design a private, yet accurate estimate, of $f(F_q(X))$. To do this, in principle we would desire $f(F_q(X))$ to be Lipschitz (with a "good" constant, say $\Lambda_f$) with respect to the Hamming distance on $X$. Indeed, with such a property the Laplace mechanism [DR14] produces an $\varepsilon$-DP estimate of $f(F_q(X))$ with error $\Lambda_f/\varepsilon$. Unfortunately, such a property cannot exist in general; as mentioned, in many cases of interest, $f$ is only known to be Lipschitz with respect to $\delta_q$ under certain conditions; for example, when $F_q(X)$ contains a ball and is contained in a ball. So, we must impose some restrictions on the input $X$.

For this reason, we design an appropriate "typical" high-probability set $\mathcal{H} \subseteq (\mathbb{R}^d)^n$, such that $f(F_q(X))$, restricted to $\mathcal{H}$ is Lipschitz, with respect to the Hamming distance on the input $X$. The properties of the typical set will allow that for all $X, Y \in \mathcal{H}$ unless the Hamming distance $d_H(X, Y)$ is "large" it holds (a) $\|f(F_q(X)) - f(F_q(Y))\|_p \leq K\delta_q(X, Y)^h$ and (b) $\delta_q(X, Y) \leq \frac{C}{Ln} d_H(X, Y)$. These results together imply $\|f(F_q(X)) - f(F_q(Y))\|_p \leq KC^h/(L^h n^h) d_H(X, Y)$. We then apply a variant of the Laplace mechanism, called the flattened Laplacian mechanism, [BCSZ18b, TVGZ20] which gives an $\varepsilon$-DP, yet accurate, estimate of $f(F_q(X))$ when $X \in \mathcal{H}$. The accuracy guarantee of the flattened Laplacian mechanism results from a careful multivariate integral calculation.

We now describe the typical set, $\mathcal{H}$. It is built as the intersection of two conditions, each one happening with high-probability. The first condition is that, in every direction, the quantiles are appropriately bounded; this is satisfied since the empirical distribution of $X$ is close to the population distribution $\mathcal{D}$ in the $\delta_q$ distance. As discussed, this condition enforces property (a) from the paragraph above. The second condition is more intricate, as it requires that, in every direction, the quantile is close, in several scales, to a non-negligible fraction of the points. This reduces the sensitivity of the quantile to the individual sample points and we use it to establish property (b). The proof that the second condition holds with high-probability *for any admissible distribution* is a combination of a net-argument and an appropriate use of the one-dimensional result in [TVGZ20, Lemma B.2].

**Extension Lemma**  Finally, a direct application of the Extension Lemma 6 extends the flattened Laplacian mechanism from the previous paragraph to a $2\varepsilon$-DP estimator on the whole $(\mathbb{R}^d)^n$, while remaining the same on inputs from $\mathcal{H}$. Since $\mathcal{H}$ happens with high probability, the result follows.

# 6 Conclusion

Our work suggests several future directions. First, our estimators come with no polynomial termination time guarantee since they are based on applying the generic Extension Lemma. Yet, in the one dimensional case, [TVGZ20] showed that the Extension Lemma can be computed in polynomial time when applied to the empirical median on an appropriate typical set. Given the analogies between our works, it is an important open problem to see if the high-dimensional estimators we propose can also be implemented in polynomial-time. Second, our work leverages classical robust convex geometric tools to construct private algorithms, which allow to demonstrate non-trivial high-dimensional phenomena. It will be interesting to see if these newly introduced tools in the privacy literature can be further used to reveal other non-trivial high-dimensional results.

# Acknowledgments

DM is supported by a Vannevar Bush Faculty Fellowship ONR-N00014-20-1-2826. IZ is supported by the Simons-NSF grant DMS-2031883 on the Theoretical Foundations of Deep Learning and the Vannevar Bush Faculty Fellowship ONR-N00014-20-1-2826.

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
