# A  Further Preliminaries

## A.1  Why Floating Bodies?

The main object of interest in the present work is the *convex floating body*. This is a well known construct in convex geometry, which takes its name from the work of Archimedes (see e.g. [Hea09]), and shares fascinating connections with non-parametric statistics. Here we further elaborate on the motivation to study floating bodies as a natural, robust high-dimensional statistical object. We refer the interested reader to the survey of Nagy et al. [NSW19] for an extensive discussion of its statistical relevance.

Let us first recall the definition of the convex floating body. Let $q \in [0, 1]$, and let $Y$ be a random variable in $\mathbb{R}$. The $q$-*quantile* of $Y$, which we denote $Q_q(Y)$, is

$$Q_q(Y) := \arg\inf\{t \in \mathbb{R} | \mathbb{P}(Y \leq t) \geq q\}.$$

That is, $Q_q(Y)$ is the infimum value satisfying $\mathbb{P}(Y \leq Q_q(Y)) \geq q$. Now, for a random vector $X$, in $\mathbb{R}^d$, the (convex) $q$-floating body, $F_q(X)$, is defined by

$$F_q(X) = \bigcap_{\theta \in \mathbb{S}^{d-1}} \left\{ x \in \mathbb{R}^d : \langle x, \theta \rangle \leq Q_q(\langle X, \theta \rangle) \right\},$$

where we denote by $\mathbb{S}^{d-1} \subset \mathbb{R}^d$ the unit Euclidean sphere, in $d$-dimensions. In other words, the convex floating body is the intersection of all halfspaces containing at least a $1 - q$ fraction of the mass of the distribution. See Figure 1 for a pictorial representation of the floating body of a data-set and [NSW19, Figures 1-4] for illustrations of floating bodies in various interesting contexts.

The floating body has the following desirable properties.

**Existence.**   The convex floating body always exists for $q \geq 1 - 1/(d+1)$. This follows from the fact that for any set $S$ of $n$ vertices in $d$ dimensions, there is some point with Tukey depth $n/(d+1)$ with respect to $S$, which in itself is a standard implication of Helly's theorem from convex analysis (e.g., [BMNS19]).

The above result holds under worst case assumptions, and can be overly pessimistic for many types of realistic data distributions (in the sense that the maximum Tukey depth guaranteed, $n/(d+1)$, depends inversely on $d$). However, for many distributions of practical interest exhibiting some sort of "niceness" properties (or "admissibility" as we say in the current work – see Definition 2, that ensures $F_q(\mathcal{D})$ contains a ball of radius $R_{\min}$ centered at $c$), much stronger guarantees are known. One example is the wide and important family of centrally symmetric log-concave distributions, which includes for example the Gaussian, uniform, and Laplace distributions, among many others. When the random vector $X$ is generated from any such admissible distribution, it suffices to take $n$ polynomial in $d$ to ensure that the empirical $q$-floating body is non-empty for any fixed $q > 1/2$ (independently of the dimension $d$); see Appendix E for more details.

**The floating body is a natural high dimensional statistical construction.**   Privately outputting descriptive statistics of a given dataset is among the most fundamental tasks in the privacy literature. Indeed, a large body of very recent work in the differential privacy literature is devoted to privately estimating quantiles in one dimension (e.g., [BAM20, TVGZ20, GJK21, KSS21, ABC22, LGG+22]), or uses the interquantile range to privately output meaningful measurements of the standard deviation of one dimensional distributions [DL09].

However, one-dimensional statistics (applied to projections of some high-dimensional data) cannot generally capture the complexity of high dimensions. Suppose, as *a running and motivating example*, that we maintain a large high-dimensional database, where each (high-dimensional) entry represents the feature vector of a single user in the database. Naturally, one might want the ability to privately generate artificial users that exhibit the "typical" behavior of actual users in the database, without compromising on users' privacy. An important application is private generation of high quality synthetic data for training machine learning models on the database, which should be accurate enough to work well on actual users, yet maintain the privacy on existing users in the database (e.g., [GAH+14, YJvdS19, BS21, MPSM21]).

Privately sampling from the floating body of the random variable $X \sim \mathcal{D}$ (for an unknown distribution $\mathcal{D}$) is arguably the most statistically principled approach to privately generating a large and diverse

yet "representative" collection of points (which is the type of access needed for the synthetic data generation application) from the unknown $\mathcal{D}$, given only sample access to it.

**Robustness.**    The existence of outliers in the data is one of the most challenging aspects to statistics and machine learning in high dimensions. One of the main difficulties is that there is no canonical definition of what constitutes an outlier in the data. The convex floating body offers a very simple, nonparametric interpretation of "central points" of the distribution: these are precisely all points in the floating body $F_q$ (where $q$ can possibly depend on the data), i.e., all points that in every direction fall in the $q$-interquantile range. This point of view affords a high-dimensional interpretation for outliers; A point $x \in \mathbb{R}^d$ is an outlier in direction $\theta \in \mathbb{S}^{d-1}$, if $\langle x, \theta \rangle > h_{F_q(\mathcal{D})}(\theta)$, where $h_{F_q(\mathcal{D})}$ is the support function of $F_q(\mathcal{D})$, as described below in (10). One appeal of using the support function, as opposed to the quantile in direction $\theta$, is that the latter only depends on a marginal, while the former depends on the joint high-dimensional distribution, and so represents a more integrative decision rule.

**Rich convex geometry foundations.**    As is demonstrated throughout the paper (and in more detail, e.g., in the survey [NSW19]), there is a very rich understanding of convex floating bodies in high dimensions from the convex geometry perspective. Indeed, objects and questions of this type have been systematically studied in the last two hundred years; the earliest modern work is Dupin's book from 1822 [Dup22]. As described in the survey, the interplay between different notions of symmetry and depth arising in convex bodies and log concave measures (which are, in many ways, the natural measure-theoretic generalization of a convex body) gives rise to deep and interesting mathematics. Thus, working with the convex body provides us access to this rich literature without the need to establish a mathematical theory from scratch.

## A.2    Omitted Useful Definitions

**Approximate Differential Privacy**    Throughout the paper we referred to a similar notion to "pure" $\varepsilon$-differential privacy (Definition 1) called "approximate" $(\varepsilon, \delta)$-differential privacy. We now formally define it.

**Definition 16.** A randomized algorithm $\mathcal{A}$ is $(\varepsilon, \delta)$-differential private if for all subsets $S \in \mathcal{F}$ of the output measurable space $(\Omega, \mathcal{F})$ and $n$-tuples of samples $X_1, X_2 \in (\mathbb{R}^d)^n$ it holds

$$\mathbb{P}\left(\mathcal{A}(X_1) \in S\right) \leq e^{\varepsilon d_H(X_1, X_2)} \mathbb{P}\left(\mathcal{A}(X_2) \in S\right) + \delta. \tag{9}$$

Clearly $(\varepsilon, \delta)$-differential privacy is a *weaker* notion to $\varepsilon$-differential privacy, in the sense that for any $\varepsilon, \delta > 0$ any $\varepsilon$-differentially private algorithm is also an $(\varepsilon, \delta)$-differentially private algorithm.

**Support function and Hausdorff distance**    Let us also introduce a useful object called the support function of a convex body. If $K \subset \mathbb{R}^d$ is a convex body its support function, $h_K : \mathbb{S}^{d-1} \to \mathbb{R}$, is defined by

$$h_K(\theta) = \max_{x \in K} \langle x, \theta \rangle. \tag{10}$$

Using the support function we can give an (alternative) functional definition of the Hausdorff distance, with the following equivalence (see [AAGM15]):

$$\delta_{\mathrm{Haus}}(K, K') = \sup_{\theta \in \mathbb{S}^{d-1}} |h_K(\theta) - h_{K'}(\theta)|. \tag{11}$$

when $K$ and $K'$ are convex bodies.

# B    Proof of Theorem 10

In this Section we establish our main meta-theorem, Theorem 10. For convenience, we first recall the statement of the theorem.

**Theorem 17** (Restated Theorem 10). *Fix $q \in (1/2, 1)$ and assume that $\mathcal{D} \in A_q(R_{\max}, R_{\min}, r, L)$. Further, for $h, K > 0$ and $A \subset \mathbb{S}^{d-1}$, let $f : \mathcal{C}_d \to \mathbb{R}^M$ be an approximate $h$-Hölder function with constant $K$, with respect to $A$. Then, there is an $\varepsilon$-differentially private algorithm $\mathcal{A}$ which for*

input $X = (X_1, \ldots, X_n)$, sampled i.i.d *from* $\mathcal{D}$, *satisfies for all* $\alpha < \min\{1, K\}^{\frac{\min\{r, R_{\min}\}}{2}}$ *that* $\mathbb{P}(\|\mathcal{A}(X) - f(F_q(\mathcal{D}))\|_p \leq \alpha) \geq 1 - \beta$, *for some* $n$ *such that*

$$n = \tilde{O}\left(\frac{K^{2/h}\left(d + \log\left(\frac{4}{\beta}\right)\right)}{\alpha^{2/h}L^2} + \frac{WK^{1/h}(\log\left(\frac{1}{\beta}\right)^{1/h} M^{1/h})}{(\varepsilon\alpha)^{1/h}L} + \frac{WM^{1/h}}{(\varepsilon\min\{r, R_{\min}\})^{1/h}L}\right),$$

*where* $W = \min\{d, \log|A|\} + \log\left(\frac{1}{\beta}\right)$.

We also recall the definition of an approximate Hölder function and fix the parameters in the definition. Thus, we say that $f : \mathcal{C}_d \to \mathbb{R}^M$ is an approximate $h$-Hölder function, with constant $K$, with respect to $A$, if,

$$\|f(F_q(X)) - f(F_q(Y))\|_p \leq K\delta_q(X, Y; A)^h,$$

whenever $F_q(X)$ contains a ball of radius $\frac{R_{\min}}{2}$, is contained in a ball of radius $R_{max} + r$, and $\delta_q(X, Y) \leq \frac{R_{\min}}{4}$.

The rest of the section, and the proof itself, is divided into three main parts. The first part is to analyze a natural non-private estimator for the task (see Section B.1). In the second part, we begin "privatizing" the non-private estimator. To do so, as described previously, we first restrict ourselves to a "typical" subset of possible inputs (described in Section B.2). On the typical subset we apply to the non-private estimator a flattened Laplace mechanism, and calculate its accuracy (see Section B.3). Finally, the third part is to extend the "restricted" estimator to the whole space of inputs while keeping the same privacy and accuracy guarantees by appropriately applying the Extension Lemma , described in Proposition 6 (see Section B.4).

## B.1 Admissibility and the Empirical Non-Private Estimator

Recall Definition 2, which introduced the minimal assumption of being an admissible distribution. The first aim of this section is to establish one desirable (yet, non-private) consequence of admissibility; the quantiles of polynomially many samples drawn from an admissible distribution are uniformly close to the quantiles of the distribution (that is, the sample is close to the distribution in the $\delta_q$ distance). In particular, for any Hölder query $f$, as in (5), simply outputting the value of $f$ on the empirical floating body produces a natural non-private estimator with desirable accuracy using polynomially many samples.

To formalize and prove this property, we shall require the following lemma from [AR20].

**Lemma 18** (Lemma 8 in [AR20]). *Let $X$ and $Y$ be two random variables with respective CDF $F_Y$ and $F_Y$. Then, for every $q \geq \frac{1}{2}$, if $b := \sup_t |F_X(x) - F_Y(t)|$ and the following conditions hold, for some $a > 0$:*

- $F_X(Q_q(X) - a) - F_X(Q_q(X)) > b$, *and*
- $F_X(Q_q(X)) - F_X(Q_q(X) - a) > b$,

*then,*

$$|Q_q(X) - Q_q(Y)| \leq a$$

The following is our non-private estimation result which applies to admissible distribution.

**Theorem 19.** *Let $X$ be a random vector in $\mathbb{R}^d$ with law in $A_q(R_{\max}, R_{\min}, r, L)$. Let $(X_i)_{i=1}^n$ be i.i.d. copies of $X$ and let $Y$ be chosen uniformly from $(X_i)_{i=1}^n$. Then, for every $\alpha, \beta > 0$ with $\alpha < \frac{r}{2}$,*

$$\mathbb{P}\left(\delta_q(X, Y) \leq \alpha\right) \geq 1 - \beta,$$

*provided that*

$$n \geq \frac{16}{\alpha^2 L^2}\left(d\log\left(\frac{16d}{\alpha^2 L^2}\right) + \log\left(\frac{4}{\beta}\right)\right).$$

*Proof.* We begin by defining the set of hyperplane threshold functions

$$\text{hyper}_d := \{f : \mathbb{R}^d \to \{0, 1\} | f(x) = \mathbf{1}(\langle x, \theta \rangle < t) \text{ for some } \theta \in \mathbb{S}^{d-1}, t \in \mathbb{R}\}.$$

For some $\alpha_1 > 0$, using standard VC arguments, as in [AR20, Theorem 7], and the fact that the VC dimension of $\text{hyper}_d$ is $d + 1$, we get

$$\mathbb{P}\left(\sup_{f \in \text{hyper}_d} |\mathbb{E}[f(X)] - \mathbb{E}[f(Y)]| \leq \alpha_1\right) \geq 1 - \beta, \tag{12}$$

whenever,

$$n \geq \frac{16}{\alpha_1^2}\left(d\log\left(\frac{16d}{\alpha_1^2}\right) + \log\left(\frac{4}{\beta}\right)\right).$$

For $\theta \in \mathbb{S}^{d-1}$, let $F_{\langle X,\theta\rangle}$ stand for CDF of $\langle X,\theta\rangle$ (and with a similar notation for $Y$). With this notation, (12) may be alternatively written as,

$$\mathbb{P}\left(\sup_{\theta,t} |F_{\langle X,\theta\rangle}(t) - F_{\langle Y,\theta\rangle}(t)| \leq \alpha_1\right) \geq 1 - \beta. \tag{13}$$

We now show that the event in (13) together with Lemma 18 and the admissibility of $X$ implies the result. Indeed, assume that $\frac{2\alpha_1}{L} < r$, fix $\theta \in \mathbb{S}^{d-1}$ and denote $X_\theta := \langle X,\theta\rangle$. In this case, since $X \in A_q(R_{\max}, R_{\min}, r, L)$, we have,

$$F_{X_\theta}\left(Q_q(X_\theta) + \frac{2\alpha_1}{L}\right) - F_{X_\theta}(Q_q(X_\theta)) = \int_{Q_q(X_\theta)}^{Q_q(X_\theta)+\frac{2\alpha_1}{L}} f_{X_\theta}(t)\mathrm{d}t \geq 2\alpha_1 > \alpha_1,$$

$$F_{X_\theta}(Q_q(X_\theta)) - F_{X_\theta}\left(Q_q(X_\theta) - \frac{2\alpha_1}{L}\right) = \int_{Q_q(X_\theta)-\frac{2\alpha_1}{L}}^{Q_q(X_\theta)} f_{X_\theta}(t)\mathrm{d}t \geq 2\alpha_1 > \alpha_1.$$

Hence, Lemma 18 implies,

$$|Q_q(X_\theta) - Q_q(Y_\theta)| \leq \frac{2\alpha_1}{L}.$$

for every $\theta \in \mathbb{S}^{d-1}$. The proof concludes by choosing $\alpha_1 = \frac{L}{2}\alpha$. for $\alpha < \frac{r}{2}$. $\square$

## B.2  The Typical Set of the Private Estimator

As mentioned in Section B.1, Theorem 19 implies the success of a non-trivial estimator for any Hölder query: Take a large sample and calculate the query on the empirical floating body. Naturally, such a procedure offers no privacy guarantees.

In order to make this algorithm private, we follow the general approach described in Section 1.3. Recall that our first step is to restrict the possible samples into "typical" ones. The "typical" samples will enjoy two important properties: (a) they are drawn with high-probability over the distribution and (b) they are not "too sensitive" to changes in a small number of sample points. These are the two properties we establish in this section. Then, using these properties in Section B.3 we construct of a "restricted private algorithm" defined only on the typical set. Finally, with the extension lemma we will produce the final private algorithm defined on every input.

**Definition of the typical set**  We now define a 'typical' subset $\mathcal{H} \subseteq (\mathbb{R}^d)^n$ of the sample space. Let $W > 1$ be a parameter, and, for a fixed direction $\theta \in \mathbb{S}^{d-1}$, define the event:

$$\mathcal{H}_W^\theta := \left\{ X \in (\mathbb{R}^d)^n : \begin{cases} \sum_{i \in [n]} \mathbf{1}\{\langle X_i, \theta\rangle - Q_q(\langle X,\theta\rangle) \in [0, \frac{\kappa W}{Ln}]\} \geq \kappa + 1 \\ \sum_{i \in [n]} \mathbf{1}\{Q_q(\langle X,\theta\rangle) - \langle X_i, \theta\rangle \in [0, \frac{\kappa W}{Ln}]\} \geq \kappa + 1 \\ \kappa \in \{1, \cdots, \frac{Lr}{2W}n\} \\ Q_q(\langle X,\theta\rangle) \in [-R_{\max} - \frac{r}{2}, R_{\max} + \frac{r}{2}] \\ \langle X_i, \theta\rangle \leq B, i \in [n] \end{cases} \right\}.$$

If $A \subset \mathbb{S}^{d-1}$ is the subset of direction we consider, *the typical set* (with respect to $A$) is defined by

$$\mathcal{H}(A) = \mathcal{H}_W(A) = \left(\bigcap_{\theta \in A} \mathcal{H}_W^\theta\right) \cap \{X \in (\mathbb{R}^d)^n | F_q(X) \text{ contains a ball of radius } R_{\min}/2\}. \tag{14}$$

We abbreviate $\mathcal{H}(\mathbb{S}^{d-1}) = \mathcal{H}$.

**Intuition behind the definition**   We now discuss some intuition. The first two conditions in $\mathcal{H}_W^\theta$ mean that, when projecting $X$ in direction $\theta$, the quantile has some fraction of points surrounding it, in different scales; in each interval $[Q_q(\langle X, \theta \rangle), Q_q(\langle X, \theta \rangle) \pm \frac{\kappa W}{Ln}]$, there are at least $\kappa + 1$ points, for every $\kappa \in \{1, \cdots, \frac{Lr}{2W} n\}$. This is the main property which guarantees the stability of the quantiles under small Hamming distance changes on the input. The third and fourth conditions in $\mathcal{H}_W^\theta$ as well as the ball containment property in $\mathcal{H}(A)$ ensure, respectively, that the quantiles and projections are bounded. Note that by Part 4 in Definition 2 we can, and do, assume

$$B = \text{poly}(n, d). \tag{15}$$

These "boundedness" properties crucially implies that the Hausdorff distance between two floating bodies is of comparable size with the "quantile" distance $\delta_q$ between them (e.g. see Lemma 8).

**High probability guarantees**   We now show that the typical set is a high-probability event, provided sufficiently many samples are drawn from an admissible distribution.

**Lemma 20.** *Suppose that the sample $X \in (\mathbb{R}^d)^n$ is drawn from an admissible distribution $\mathcal{D} \in A_q(R_{\max}, R_{\min}, r, L)$ and that $W \le e^n$ is large enough. Then, for any $A \subset \mathbb{S}^{d-1}$*

$$\mathbb{P}[\mathcal{H}_W(A)] \ge 1 - We^{-\tilde{\Theta}(W + \min(\log(|A|), d))} - e^{-\tilde{\Theta}(L^2 r^2 n + \min(\log(|A|), d))},$$

*whenever*

$$n = \tilde{\Omega}\left( \frac{d}{\min\{R_{\min}, r\}^2 L^2} \right).$$

*In particular, for any $\beta > 0$, we can ensure,*

$$\mathbb{P}[\mathcal{H}_W(A)] \ge 1 - \beta,$$

*for some,*

$$n = \tilde{O}\left( \frac{d}{\min\{R_{\min}, r\}^2 L^2} + \log\left( \frac{1}{\beta} \right) \right) \text{ and } W = \tilde{O}\left( \min\left( \log(|A|), d \right) + \log\left( \frac{1}{\beta} \right) \right).$$

The proof, which is deferred to the following section, is an outcome of combining the one-dimensional Lemma B.3. of [TVGZ20] and an appropriate covering argument.

**Low Hamming distance sensitivity**   Our next task is to show that typical samples produce empirical quantiles which are not very sensitive to individual sample points. Note that if one changes all $n$ sample points the quantile can take arbitrary values in $[-R_{\min} - r/2, R_{\min} + r/2]$, given that they only need to be realized from input in the typical set. Our next result shows when a fraction of the sample points change, the typical set guarantees the quantiles remain stable.

**Lemma 21.** *Let $A \subset \mathbb{S}^{d-1}$ and suppose $X, Y \in \mathcal{H}_W(A)$ with Hamming distance $d_H(X, Y) \le \frac{Lr}{2W} n$. Then,*

$$\delta_q(X, Y; A) \le \frac{2W}{Ln} d_H(X, Y).$$

*In particular,*

$$\frac{Ln}{2W} \min\{\delta_q(X, Y; A), r\} \le d_H(X, Y).$$

The proof is also deferred to the following section.

### B.2.1   Proofs of Lemma 20 and Lemma 21

*Proof of Lemma 20.* First, according to Definition 2, we may assume that, that when $B$ satisfies (15) with a large enough degree,

$$\mathbb{P}\left( \max_i \|X\|_2 > B \right) \le e^{-n}.$$

Moreover, by taking $\alpha = \frac{\min\{R_{\min}, r\}}{2}$ and $\beta = e^{-\Theta(n)}$ in Theorem 19, we can see that when

$$n = \tilde{\Omega}\left( \frac{d}{\min\{R_{\min}, r\}^2 L^2} \right),$$

we have

$$\mathbb{P}\left(\delta_q(X, \mathcal{D}) > \frac{\min\{R_{\min}, r\}}{2}\right) \leq e^{-\Theta(n)}. \tag{16}$$

In particular, coupled with Definition 7, this shows that there exists $c \in \mathbb{R}^d$, such that,

$$\mathbb{P}\left(\forall \theta \in \mathbb{S}^{d-1}, |Q_q(\langle X, \theta \rangle) - \langle c, \theta \rangle| \leq \frac{R_{\min}}{2} \text{ and } |Q_q(\langle X, \theta \rangle)| \leq R_{\max} + \frac{r}{2}\right) \leq e^{-\Theta(n)}.$$

Thus, since our claim probabilities are of larger order than $e^{-n}$, the rest of the proved is focused on bounding the probabilities of the events

$$\left\{\sum_{i \in [n]} \mathbf{1}\left\{\langle X_i, \theta \rangle - Q_q(\langle X, \theta \rangle) \in \left[0, \frac{\kappa W}{Ln}\right]\right\} \geq \kappa + 1\right\},$$

$$\left\{\sum_{i \in [n]} \mathbf{1}\left\{Q_q(\langle X, \theta \rangle) - \langle X_i, \theta \rangle \in \left[0, \frac{\kappa W}{Ln}\right]\right\} \geq \kappa + 1\right\},$$

for $\kappa \in \{1, \cdots, \frac{Lr}{2W}n\}$.

We now claim that, for a fixed $\theta \in \mathbb{S}^{d-1}$,

$$\mathbb{P}\left[\mathcal{H}_W^\theta\right] \geq 1 - We^{-\Theta(W)} - e^{-\Theta(L^2 r^2 n)}. \tag{17}$$

Indeed, this is a consequence of [TVGZ20, Lemma B.2]. Note that Definition 2 implies that the marginal of $\mathcal{D}$, in direction $\theta$, which we denote as $\mathcal{D}_\theta$, is an admissible distribution around $Q_q(\mathcal{D}_\theta)$, in the sense of [TVGZ20, Assumption 1.2], and (17) follows, mutatis-mutandis, from the proof of [TVGZ20, Lemma B.2][5].

If $|A| < e^d$, then, with a union-bound

$$\mathbb{P}(\mathcal{H}_W(A)) \geq 1 - |A|\left(We^{-\Theta(W)} + e^{-\Theta(L^2 r^2 n)}\right)$$

$$= 1 - \left(We^{-\Theta(W + \log(|A|))} + e^{-\Theta(L^2 r^2 n) + \log(|A|)}\right).$$

When $|A| \geq e^d$, we will use the inclusion $\mathcal{H}_W(A) \subset \mathcal{H}_W$, and prove the result uniformly on the sphere. For this, fix $\gamma := \frac{W}{4Bn}$ and let $N_\gamma \subset \mathbb{S}^{d-1}$ be an $\gamma$-net. Standard arguments show that one can ensure $|N_\gamma| \leq \left(\frac{3}{\gamma}\right)^d = e^{d\log\left(\frac{3}{\gamma}\right)}$. Denote $\mathcal{H}_W^\gamma := \bigcap_{\theta \in N_\gamma} \mathcal{H}_W$. A union bound over $N_\gamma$ shows,

$$\mathbb{P}\left[\mathcal{H}_W^\gamma\right] \geq 1 - \left(We^{-\Theta\left(W + d\log\left(\frac{3}{\gamma}\right)\right)} + e^{-\tilde{\Theta}(L^2 r^2 n) + d\log\left(\frac{3}{\gamma}\right)}\right). \tag{18}$$

Now, assume $\mathcal{H}_W^\gamma$ holds, and let $\theta \in \mathbb{S}^{d-1}$ with $\theta' \in N_\gamma$ such that, $\|\theta - \theta'\| \leq \gamma$. In this case, since the $\|X_i\| \leq B$, for every $i = 1, \ldots, n$,

$$|\langle X_i, \theta \rangle - \langle X_i, \theta' \rangle| \leq \|X_i\|\|\theta - \theta'\| \leq B\gamma \leq \frac{W}{4n}.$$

This implies the following bound on the infinity Wasserstein distance: $W_\infty\left(\langle X_i, \theta \rangle, \langle X_i, \theta' \rangle\right) \leq \frac{W}{4n}$, which in turn implies $|Q_q(\langle X, \theta \rangle) - Q_q(\langle X, \theta' \rangle)| \leq \frac{W}{4n}$ (the reader is referred to Section 2.3 in [Rac85], and the discussion following Equation 2.14, for more details). In particular, for every $i$,

$$|\langle X_i, \theta \rangle - Q_q(\langle X, \theta \rangle)| \leq \frac{W}{2n} + |\langle X_i, \theta' \rangle - Q_q(\langle X, \theta' \rangle)|.$$

Thus, we have proved the implication $\mathcal{H}_W^\theta \subset \mathcal{H}_{\frac{W}{4}}^{\theta'}$. By (18),

$$\mathbb{P}\left(\mathcal{H}_{\frac{W}{4}}\right) \geq \mathbb{P}\left(\mathcal{H}_W^\gamma\right) \geq 1 - \left(We^{-\Theta\left(W + d\log\left(\frac{3}{\gamma}\right)\right)} + e^{-\tilde{\Theta}(L^2 r^2 n) + d\log\left(\frac{3}{\gamma}\right)}\right)$$

$$\geq 1 - \left(We^{-\Theta\left(W + d\log\left(\frac{12Bn}{W}\right)\right)} + e^{-\tilde{\Theta}(L^2 r^2 n) + d\log\left(\frac{12Bn}{W}\right)}\right).$$

By (15), $\log\left(\frac{12Bn}{W}\right) = O(\log(n))$, hence we subsume it in the $\tilde{\Theta}$ notation, completing the proof. $\square$

---

[5]We set $T = 1$ in [TVGZ20, Lemma B.2]

*Proof of Lemma 21.* It suffices to show for every $\theta \in A$,

$$|Q_q(\langle Y, \theta \rangle) - Q_q(\langle X, \theta \rangle)| \leq \frac{W}{2Ln} d_H(X, Y). \tag{19}$$

We again employ Lemma 18. Let $F_\theta$ stand for the CDF of the empirical distribution of $\langle Y_i, \theta \rangle, i = 1, \ldots, n$ and $G_\theta$ for the CDF of the empirical distribution of $\langle X_i, \theta \rangle, i = 1, \ldots, n$. It clearly holds that

$$\|F - G\|_\infty \leq \frac{d_H(X, Y)}{n}.$$

Now, since $X \in \mathcal{H}(A)$ and $d_H(X, Y) \leq \frac{Lr}{2W} n$, if $a = \frac{W}{2Ln} d_H(X, Y)$, it holds that,

$$G_\theta(Q_q(\langle X, \theta \rangle) + a) - G_\theta(Q_q(\langle X, \theta \rangle)) > \frac{d_H(X, Y)}{n}.$$

Then (19) follows from Lemma 18. The second part follows from rearranging the terms. Indeed, note that if $r < \delta_q(X, Y; A)$, then,

$$\frac{Ln}{2W} \min\{\delta_q(X, Y; A), r\} = \frac{Lr}{2W} n \leq d_H(X, Y).$$

$\square$

## B.3 The Restricted Private Algorithm: Construction and Analysis

The aim of this section is to construct a private algorithm and prove that it satisfies the conclusion of our meta-theorem, Theorem 10. The construction of the algorithm is naturally based on the construction of the typical set in Section B.2.

### B.3.1 Definition of the Restricted Private Algorithm on the Typical Set

To prepare the proof, we define a randomized algorithm $\hat{\mathcal{A}}$ on inputs from the typical set $\mathcal{H} = \mathcal{H}_W(A)$, as defined in Section B.2.

On input $X \in \mathcal{H}$, the density of the algorithm is given by,

$$f_{\hat{\mathcal{A}}(X)}(t) = \frac{1}{\hat{Z}_X} \exp\left(-\frac{\varepsilon}{4} \min\left\{\left(\frac{Ln}{2W}\right)^h K^{-1}\|t - f(F_q(X))\|_p, \left(\frac{Ln \min\{r, R_{\min}\}}{8W}\right)^h\right\}\right), \tag{20}$$

on the region $\{\|t\|_p \leq 2K(R_{\max} + r/2)\}$. The density is 0 outside of this region. The normalizing constant, $\hat{Z}_X$, is given by,

$$\hat{Z}_X = \int\limits_{\|t\|_p \leq 2K(R_{\max}+r/2)} \exp\left(-\frac{\varepsilon}{4} \min\left\{\left(\frac{Ln}{2W}\right)^h K^{-1}\|t - f(F_q(X))\|_p, \left(\frac{Ln \min\{r, R_{\min}\}}{8W}\right)^h\right\}\right) \mathrm{d}t. \tag{21}$$

We call this distribution a "flattened" Laplacian mechanism, similar to [TVGZ20].

### B.3.2 Privacy Guarantees of the Restricted Algorithm

Our first result is to show that $\hat{A}$ is an $\varepsilon$-differentially private algorithm we shall require the following lemma.

**Lemma 22.** *Suppose $n > \frac{2W}{L}$. The algorithm $\hat{\mathcal{A}}$, defined on $\mathcal{H}(A)$, is $\frac{\varepsilon}{2}$-differentially private.*

*Proof.* It will be enough to show that, whenever $X, Y \in \mathcal{H}(A)$,

$$\frac{f_{\hat{\mathcal{A}}(X)}(t)}{f_{\hat{\mathcal{A}}(Y)}(t)} \leq e^{\frac{\varepsilon}{2} d_H(X, Y)}, \tag{22}$$

for every $t \in \mathbb{R}^M$, $\|t\|_p \le 2K(R + r/2)$. We first observe that, by applying the reverse triangle inequality,

$$\frac{\exp\left(-\frac{\varepsilon}{4K}\min\left\{(\frac{Ln}{2W})^h K^{-1}\|t - f(F_q(X))\|_p, (\frac{Ln\min\{r, R_{\min}\}}{8W})^h\right\}\right)}{\exp\left(-\frac{\varepsilon}{4K}\min\left\{(\frac{Ln}{2W})^h K^{-1}\|t - f(F_q(Y))\|_p, (\frac{Ln\min\{r, R_{\min}\}}{8W})^h\right\}\right)}$$

$$= \exp\left(-\frac{\varepsilon}{4K}\left(\min\left\{(\frac{Ln}{2W})^h K^{-1}\|t - f(F_q(X))\|_p, (\frac{Ln\min\{r, R_{\min}\}}{8W})^h\right\}\right.\right.$$

$$\left.\left. - \min\left\{(\frac{Ln}{2W})^h K^{-1}\|t - f(F_q(Y))\|_p, (\frac{Ln\min\{r, R_{\min}\}}{8W})^h\right\}\right)\right)$$

$$\le \exp\left(\frac{\varepsilon}{4K}\min\left\{(\frac{Ln}{2W})^h K^{-1}\|f(F_q(X)) - f(F_q(Y))\|_p, (\frac{Ln\min\{r, R_{\min}\}}{8W})^h\right\}\right)$$

$$\le \exp\left(\frac{\varepsilon}{4}\min\left\{(\frac{Ln}{2W}\delta_q(X, Y; A))^h, (\frac{Ln\min\{r, R_{\min}\}}{8W})^h\right\}\right)$$

$$\le \exp\left(\frac{\varepsilon}{4}\min\left\{(\frac{Ln}{2W}\delta_q(X, Y; A))^h, (\frac{Lnr}{8W})^h\right\}\right)$$

$$\le \exp\left(\frac{\varepsilon}{4}d_H(X, Y)^h\right) \le \exp\left(\frac{\varepsilon}{4}d_H(X, Y)\right),$$

where the second inequality is the Hölder property (5) using that $\delta_q(X, Y; A) \le R_{\min}/4$, and the second to last inequality is Lemma 21. The last inequality is a simple consequence of the fact that the Hamming distance takes non-negative integer values. Since the above inequality holds for every $t \in \mathbb{R}^M$, we may integrate it to obtain,

$$\frac{\hat{Z}_X}{\hat{Z}_Y} \le e^{\frac{\varepsilon}{4}d_H(X, Y)}.$$

Combining the two estimates gives (22). $\qquad\square$

### B.3.3 The Accuracy of the Restricted Algorithm

The analysis of the accuracy of $\hat{\mathcal{A}}$, on the typical set, will be preformed in two steps. First, we will bound the normalizing constant in (21) from below, then we shall establish that the integral over (20) is small. We record the following elementary calculation that will facilitate the coming calculations.

**Lemma 23.** *Let* $k \in \mathbb{N}$, *and set* $g_k(x) = (x^k + kx^{k-1} + k(k-1)x^{k-2} + \ldots + k!)$. *Then, for any* $0 < a < b$ *it holds,*

$$\int_a^b t^k e^{-t}\mathrm{d}t = g_k(a)e^{-a} - g_k(b)e^{-b}.$$

*Proof.* We prove the claim by induction. When $k = 0$, $g_0 \equiv 1$, and the base case follows. Otherwise, use integration by parts,

$$\int_a^b t^k e^{-t}\mathrm{d}t = -t^k e^{-t}\Big|_a^b + k\int_a^b t^{k-1}e^{-t}\mathrm{d}t = a^k e^{-a} - b^k e^{-b} + kg_{k-1}(a)e^{-a} - kg_{k-1}(b)e^{-b}$$

$$= (a^k + kg_{k-1}(a))e^{-a} - (b^k + kg_{k-1}(b))e^{-b} = g_k(a)e^{-a} - g_k(b)e^{-b}.$$

The last identity uses the observation that $g_k(x) = x^k + kg_{k-1}(x)$. $\qquad\square$

**Lemma 24.** *Suppose* $W > 1$ *and let* $X \in \mathcal{H}_W(A)$. *Then for any* $L, R_{\max}, R_{\min}, r > 0$ *and* $\alpha \in (0, \min\{r, R_{\min}\})$, $\beta \in (0, 1)$ *for some*

$$n = O\left(WK^{1/h}\frac{(\log\frac{1}{\beta})^{1/h} + (M\log M)^{1/h}}{(\varepsilon\alpha)^{1/h}L} + W\frac{M^{1/h}(\log\left(\frac{R_{\max}+r}{\alpha} + 1\right))^{1/h}}{(\varepsilon\min\{r, R_{\min}\})^{1/h}L}\right).$$

*it holds*

$$\mathbb{P}[\|\hat{\mathcal{A}}(X) - f(F_q(X))\|_p \geq \alpha] \leq \beta,$$

*where the probability is with respect to the randomness of the algorithm $\hat{\mathcal{A}}$.*

*Proof.* We prove the claim under the assumption that $K = 1$, in (5), the general case follows by re-scaling by $K$ of the target error $\alpha$ and the output of the flattened Laplace mechanism. Moreover, to slightly ease notation we set $r' = \min\{r, R_{\min}\}$ as the variable $R_{\min}$ appears in the definition of the mechanism only together with $r$ via the $\min$ operator. The final result simply need to replace $r'$ with $\min\{r, R_{\min}\}$. We also denote $R_{\max}$ simply by $R$.

In our calculations we shall use the following change of coordinates: for any function $h : \mathbb{R}_+ \to \mathbb{R}$,

$$\int_{\mathbb{R}^M} h(\|t\|_p)\mathrm{d}t = \omega_{M,p} \int_0^\infty x^{M-1}h(x)\mathrm{d}x, \tag{23}$$

where $\omega_{M,p} > 0$ is some explicit constant (see e.g. [BGMN05, Page 5]). Moreover, by combining the Hölder property of $f$ and that the zero data-set is inside the typical set we have

$$\|f(F_q(X))\|_p \leq \delta_q(X, 0; A) \leq (R + r/2)^h \leq R + r/2,$$

assuming without loss of generality $R > 1$.

**Step 1**: By switching to polar coordinates, as in (23), some elementary algebra and since $r' > r$, we have,

$$\hat{Z}_X = \int_{\|t\|_p \leq 2(R+r/2)} \exp\left(-\frac{\varepsilon}{4} \min\left\{\left(\frac{Ln}{2W}\right)^h \|t - f(F_q(X))\|_p, \left(\frac{Lnr'}{8W}\right)^h\right\}\right) \mathrm{d}t$$

$$\geq \int_{\|t\|_p \leq R+r/2} \exp\left(-\frac{\varepsilon}{4} \min\left\{\left(\frac{Ln}{2W}\right)^h \|t\|_p, \left(\frac{Lnr'}{8W}\right)^h\right\}\right) \mathrm{d}t$$

$$\geq \omega_{M,p} \int_0^{R+r/2} x^{M-1} \exp\left(-\frac{\varepsilon}{4} \min\left\{\left(\frac{Ln}{2W}\right)^h x, \left(\frac{Lnr'}{8W}\right)^h\right\}\right) \mathrm{d}x$$

$$\geq \omega_{M,p} \int_0^{r'/2} x^{M-1} \exp\left(-\frac{\varepsilon}{4} \cdot \left(\frac{Ln}{W}\right)^h x\right) \mathrm{d}x$$

$$\geq \omega_{M,p} \left(\frac{4W^h}{(nL)^h\varepsilon}\right)^{M-1} \left((M-1)! - g_{M-1}\left(\frac{(nL)^h\varepsilon r'}{8W^h}\right) \exp\left(-\frac{(nL)^h\varepsilon r'}{8W^h}\right)\right),$$

where the last inequality is Lemma 23. Now, assuming $n = \omega\left(\frac{W(M\log M)^{1/h}}{(\varepsilon r')^{1/h}L}\right)$ we have

$$g_{M-1}\left(\frac{(nL)^h\varepsilon r'}{8W^h}\right) \exp\left(-\frac{(nL)^h\varepsilon r'}{8W^h}\right) \leq M\left(\frac{(nL)^h\varepsilon r'}{8W^h}\right)^{M-1} \exp\left(-\frac{(nL)^h\varepsilon r'}{8W^h}\right) < \frac{(M-1)!}{2},$$

which implies

$$\hat{Z}_X \geq \omega_{M,p} \left(\frac{4W^h}{(nL)^h\varepsilon}\right)^{M-1} \frac{(M-1)!}{2}. \tag{24}$$

**Step 2**: By applying (23), similarly to Step 1,

$$\mathbb{P}[\|\hat{\mathcal{A}}(X) - f(F_q(X))\|_p \geq \alpha]$$

$$\leq \frac{\omega_{M,p}}{\hat{Z}_X} \int_{\alpha}^{3(R+r/2)} x^{M-1} \exp\left(-\frac{\varepsilon}{4} \min\left\{\left(\frac{Ln}{2W}\right)^h x, \left(\frac{Lnr'}{8W}\right)^h\right\}\right) \mathrm{d}x$$

$$\leq \frac{\omega_{M,p}}{\hat{Z}_X} \left(\int_{\alpha}^{+\infty} x^{M-1} \exp\left(-\frac{\varepsilon}{4}\left(\frac{Ln}{2W}\right)^h x\right) \mathrm{d}x + \right.$$

$$\left. \int_{r'/2^h}^{3(R+r/2)} x^{M-1} \exp\left(-\frac{\varepsilon}{4}\left(\frac{Lnr'}{8W}\right)^h\right) \mathrm{d}x\right)$$

$$\leq \frac{\omega_{M,p}}{\hat{Z}_X} \left(\left(\frac{4W^h}{(nL)^h \varepsilon}\right)^{M-1} g_{M-1}\left(\frac{\varepsilon}{4}\left(\frac{nL}{2W}\right)^h \alpha\right) \exp\left(-\frac{\varepsilon}{4}\left(\frac{nL}{2W}\right)^h \alpha\right)\right.$$

$$\left. + (3(R+r/2))^M \exp\left(-\frac{\varepsilon}{4}\left(\frac{Lnr'}{8W}\right)^h\right)\right)$$

Hence we conclude for all $X \in \mathcal{H}_W(A)$,

$$\mathbb{P}[\|\hat{\mathcal{A}}(X) - f(F_q(X))\|_p \geq \alpha]$$

$$\leq 4\left(\frac{g_{M-1}\left(\frac{\varepsilon}{4}\left(\frac{nL}{2W}\right)^h \alpha\right)}{(M-1)!} \exp\left(-\frac{\varepsilon}{4}\left(\frac{nL}{2W}\right)^h \alpha\right) + \right.$$

$$\left. (3(R+r/2)))^M \left(\frac{(nL)^h \varepsilon}{4W^h}\right)^{M-1} \exp\left(-\frac{\varepsilon}{4}\left(\frac{Lnr'}{8W}\right)^h\right)\right)$$

From this and an elementary asymptotic calculation, we conclude that for $W > 1$ when

$$n = \Omega\left(W \frac{(\log\frac{1}{\beta})^{1/h} + (M\log M)^{1/h}}{(\varepsilon\alpha)^{1/h} L} + W \frac{(M\log(((\frac{R}{r'} + \frac{r}{r'} + 1)M)))^{1/h} + (\log\frac{1}{\beta})^{1/h}}{(\varepsilon r')^{1/h} L}\right)$$

it holds for all $X \in \mathcal{H}$, $\mathbb{P}[\|\hat{\mathcal{A}}(X) - f(F_q(X))\|_p \geq \alpha] \leq \beta$. Since $\alpha \leq r'$, the above sample complexity bound simplifies to

$$n = \Omega\left(W \frac{(\log\frac{1}{\beta})^{1/h} + (M\log M)^{1/h}}{(\varepsilon\alpha)^{1/h} L} + W \frac{M^{1/h}(\log\left(\frac{R+r}{\alpha} + 1\right))^{1/h}}{(\varepsilon r')^{1/h} L}\right).$$

The proof of the Lemma is complete. $\square$

## B.4 Putting it Together: The Extension Lemma and the Proof of Theorem 10

In this Section we put everything together and conclude Theorem 10 from an appropriate use of the Extension Lemma.

*Proof of Theorem 10.* We use the Extension Lemma, as in Proposition 6, to extend $\hat{\mathcal{A}}$ to the entire space of inputs, $(\mathbb{R}^d)^n$, endowed with the Hamming distance (for more details on this step we direct the reader to Section F). Call the extension $\mathcal{A}$, and note that $\mathcal{A}$ is $\varepsilon$-differentially private. Indeed, by Lemma 22, $\hat{\mathcal{A}}$ is $\frac{\varepsilon}{2}$-differentially private, and Proposition 6 implies the required privacy guarantees for $\mathcal{A}$. Moreover, for all $X \in \mathcal{H}_W(A)$, $\hat{\mathcal{A}}(X) \overset{\text{law}}{=} \mathcal{A}(X)$. Thus, we are left with addressing the accuracy of $\mathcal{A}$.

First, we note that by the assumptions of the theorem

$$W = \Omega\left(\min\{\log(|A|), d\log(B/W)\} + \log\left(\frac{1}{\beta}\right)\right),$$

and

$$n = \Omega\left(\frac{d}{L^2 \min\{R_{\min}, r\}^2} + \log\left(\frac{1}{\beta}\right)\right).$$

Therefore, by Lemma 20, with probability $1 - \frac{\beta}{2}$, we have $X \in \mathcal{H}(A)$, and so $\mathcal{A}(X), \hat{\mathcal{A}}(X)$ follow the same distribution.

Hence, under the event $X \in \mathcal{H}(A)$, by Lemma 24, we have that for some

$$n = O\left(WK^{1/h}\frac{(\log\frac{1}{\beta})^{1/h} + (M\log M)^{1/h}}{(\varepsilon\alpha)^{1/h}L} + W\frac{M^{1/h}(\log\left(\frac{R_{\max}+r}{\alpha} + 1\right))^{1/h}}{(\varepsilon\min\{r, R_{\min}\})^{1/h}L} + \frac{d}{L^2\min\{R_{\min}, r\}^2}\right),$$

with probability $1 - \frac{\beta}{2}$, it holds

$$\|\mathcal{A}(X) - f(F_q(X))\|_p \leq \frac{\alpha}{2}.$$

Since $\alpha < KR_{\min}/2$, by Theorem 19, there is some

$$n = O\left(\frac{K^{2/h}}{\alpha^{2/h}L^2}\left(d\log\left(\frac{16dK^{2/h}}{\alpha^{2/h}L^2}\right) + \log\left(\frac{4}{\beta}\right)\right)\right),$$

such that, with probability $1 - \frac{\beta}{2}$, it holds that

$$\delta_q(X, \mathcal{D}; A) \leq \left(\frac{\alpha}{2K}\right)^{1/h}.$$

Using the triangle inequality and then the Hölder property, (5), we have

$$\|\mathcal{A}(X) - f(F_q(\mathcal{D}))\|_p \leq \|\mathcal{A}(X) - f(F_q(X))\|_p + \|f(F_q(X)) - f(F_q(\mathcal{D}))\|_p$$
$$\leq \frac{\alpha}{2} + K\delta_q(X, \mathcal{D}; A)^h \leq \alpha.$$

A union bound shows that the probability of the event above is at least $1 - \beta$. The result then follows. □

## C  The Steiner Point: Proof of Lemma 12

In this section we prove Lemma 12, the stepping stone from Theorem 10 to Corollary 13. Recall that, if $K \subset \mathbb{R}^d$ is a convex body, its Steiner point is given by,

$$S(K) := d\int_{\mathbb{S}^{d-1}} \theta h_K(\theta)d\sigma,$$

where $\sigma$ is the uniform probability measure measure on $\mathbb{S}^{d-1}$, and $h_K$ is the support function of $K$, as in (10).

*Proof of Lemma 12.* We first show that, for every convex body $K$, $S(K) \in K$. Indeed, define $f_K(\theta) = \arg\max_{x \in K}\langle x, \theta\rangle$. Observe that $\nabla h_K = f_K$. Hence, a straightforward application of the divergence theorem (see [PaY89, Chapter 6]) gives:

$$S(K) := d\int_{\mathbb{S}^{d-1}} \theta h_K(\theta)d\sigma = \int_{\mathbb{S}^{d-1}} f_K(\theta)d\sigma,$$

So, $S(K)$ is a convex combination of $f_K(\theta)$. By definition, for every $\theta \in \mathbb{S}^{d-1}$, $f_K(\theta) \in K$ which implies, through convexity, $S(K) \in K$. To prove that $S$ is Lipschitz, let $K' \subset \mathbb{R}^d$ be any other convex body. We have

$$\|S(K) - S(K')\|_2 = \sup_{u \in \mathbb{S}^{d-1}}\langle u, S(K) - S(K')\rangle \leq d\int_{\mathbb{S}^{d-1}} |\langle u, \theta\rangle||h_K(\theta) - h_{K'}(\theta)|d\sigma$$

$$\leq \delta_{\mathrm{Haus}}(K, K')d\int_{\mathbb{S}^{d-1}} |\langle u, \theta\rangle|d\sigma \leq \delta_{\mathrm{Haus}}(K, K')d\sqrt{\int_{\mathbb{S}^{d-1}} |\langle u, \theta\rangle|^2 d\sigma}$$

$$= \sqrt{d}\delta_{\mathrm{Haus}}(K, K').$$

The second inequality uses (11) and the third is Jensen's inequality. The last identity uses the well-known fact, obtained through symmetry, that the averaged square of a coordinate on $\mathbb{S}^{d-1}$ is $\frac{1}{d}$. Finally, let $X, X' \in (\mathbb{R}^d)^n$ and assume that $F_q(X)$ contains a ball of radius $R_{\min}/2$, is contained in a ball of radius $R_{\max} + r$. centered at the origin, and that $\delta_q(X, X') \leq \frac{R_{\min}}{4}$.

This allows us to invoke Lemma 8, which, when coupled with the above bound, yields

$$\|S(F_q(X)) - S(F_q(X'))\|_2 \leq \sqrt{d}\delta_{\text{Haus}}(F_q(X), F_q(X')) \leq 6\sqrt{d}\frac{R_{\max} + r}{R_{\min}}\delta_q(X, X')$$

and concludes the proof. $\qquad\square$

# D  Private Projection and Sampling

In this section we prove Corollary 15, restated below for convenience.

**Corollary 25** (Corollary 15, restated). *Let $\mathcal{D} \in A_q(R_{\max}, R_{\min}, r, L)$ be an admissible measure on $\mathbb{R}^d$ and let $U_q$ be a random vector which is uniform on $F_q(\mathcal{D})$. Assume that $F_q(\mathcal{D})$ contains a ball of radius $R_{\min}$ around the Steiner point $S(K)$. Then, there is an $\varepsilon$-differentially private algorithm $\mathcal{A}$ which for input $X = (X_1, \ldots, X_n)$, sampled i.i.d from $\mathcal{D}$, satisfies for all $\alpha < \frac{\min\{1, r, R_{\min}\}}{2}$,*

$$\frac{1}{d}W_2^2(\mathcal{A}(X), U_q) \leq \alpha,$$

*for some*

$$n = \tilde{O}\left(\frac{\text{poly}(R_{\max} + r + 1)}{\text{poly}(R_{\min})}\left(\frac{d^2}{\alpha^{14}L^2} + \frac{d^4}{\varepsilon^2\alpha^8 L} + \frac{d^4}{\varepsilon^2\alpha^2\min\{r, R_{\min}\}^2 L}\right)\right).$$

## D.1  Proof Sketch

We start with outlining the steps we follow to establish Corollary 15.

- We first employ a non-private algorithm from the sampling literature [Leh21], restated below in Theorem 26, which produces an approximate uniform sample from a convex body given access only to a) the Steiner point of the body and b) projection operator onto the body.

- Our next step is to establish that the sampling algorithm is robust to a certain amount of noise. That is, given access to an approximate version of the Steiner point and the projection operator, the non-private algorithm still produces an approximate uniform point from the convex body of interest.

- Our final step specializes to privately sampling from the convex body of interest, the floating body of a distribution, where we remind the reader that we are only given samples from the distribution. Using the previous steps, this part is proven by appropriate applications of our meta-theorem, Theorem 10. We show that Theorem 10 implies that differentially private estimators can achieve the desired approximation guarantees, both when applied to the Steiner point (as proven in Corollary 13) and the projection operator (see Corollary 31 below).

## D.2  Background in Wasserstein Distances

We start with some background material on the $W_p$ distances.

First, for $p \geq 1$, we define the Wasserstein distance between two random vectors $X, Y \in \mathbb{R}^d$, as

$$W_p(X, Y) := \inf_{(X,Y)} \left(\mathbb{E}\left[\|X - Y\|_2^p\right]\right)^{\frac{1}{p}},$$

where the infimum is taken over all coupling of $X$ and $Y$; that is, random vectors in $\mathbb{R}^{2d}$ whose marginal on the first (resp. last) $d$ coordinates has the same law as $X$ (resp. $Y$). The Wasserstein distance turns out to be a metric which metrizes weak convergence and convergence of the first $p$ moments, see [Rac85] for further details. In this work we are mainly interested in the quadratic Wasserstein distance $W_2$. However, note that bounds on $W_2$ gives the same guarantees for $W_p$, when $p \leq 2$. Indeed, by Jensen's inequality,

$$W_p \leq W_{p'} \text{ whenever } p \leq p'.$$

### D.3 A Non-private Sampling Algorithm

If $K \subset \mathbb{R}^d$ is a convex body, we utilize the following, non-private, sampling algorithm with guarantees in $W_2$. Set $\eta > 0$ and consider the discretized Langevin process:

$$X_{t+1} = P_K(X_t + \eta g_t), \quad X_0 = S(K), \tag{25}$$

where $\{g_t\}_{t \geq 0}$ are *i.i.d.* standard Gaussians, and $S(K)$ is the Steiner point of $K$, as in (7). The following result holds.

**Theorem 26** ([Leh21, Theorem 2]). *Suppose that $K$ contains a ball of radius $R_{\min}$, centered at $S(K)$, and is contained in a ball of radius $R_{\max}$. Then, if, for some $\alpha > 0$, we take $\eta = \tilde{\Theta}\left(\frac{R_{\min}^2}{(R_{\max}+1)^4} \frac{\alpha^2}{d}\right)$ and $k = \tilde{\Theta}\left(\left(\frac{R_{\max}+1)^6}{R_{\min}^2} \frac{d}{\alpha^2}\right)$, the following bound holds:*

$$\frac{1}{d} W_2^2(X_k, U_K) \leq \alpha,$$

*where $U_K$ is a random vector, uniformly distributed over $K$.*

### D.4 Noise Robustness of the Non-private Algorithm

Theorem 26 requires *exact access* to the projection operator and to the Steiner point. To allow some uncertainty we now define the notion of a noisy projection oracle.

**Definition 27** (Noisy oracles for $K$). We say that the random function $\tilde{P}_K$ is an $(\alpha, \beta, R)$-noisy projection oracle for $K$, if the following two conditions are met, for every $x \in \mathbb{R}^d$,

1. $\|P_K(x) - \tilde{P}_K(x)\|_2 < R$, almost surely.

2. $\mathbb{P}\left(\|P_K(x) - \tilde{P}_K(x)\|_2 < \alpha\right) > 1 - \beta$.

By extension we say that the random point $\tilde{S}(K)$ is a $(\alpha, \beta, R)$-noisy oracle for the Steiner point, if it satisfies the same conditions above with respect to the $S(K)$.

Given noisy oracles for $K$, we can define a noisy version of the Langevin process. In order to take advantage of Corollary 31, we shall also needs the projected quantities to have bounded norm. Thus, for $k, \alpha > 0$, let $\{\tilde{g}_t\}_{t \geq 0}$ be *i.i.d* random vectors with the law of the standard Gaussian, conditioned to have norm at most $\sqrt{d} \log(dk)$. We then define the noisy Langevin process as

$$\tilde{X}_{t+1} = \tilde{P}_K(\tilde{X}_t + \eta \tilde{g}_t), \quad \tilde{X}_0 = \tilde{S}(K). \tag{26}$$

We now show that the noisy and noiseless versions cannot differ by much.

**Lemma 28.** *Fix $k \in \mathbb{N}$ and suppose that $\tilde{P}_K$ and $\tilde{S}(K)$ are $(\alpha, \beta, R)$-noisy oracles for $P_K$ and $S(K)$, and that $K$ is contained in a ball or radius $R_{\max}$. Then, for every $t \leq k$, there is a coupling of $X_t$ and $\tilde{X}_t$, such that,*

$$\mathbb{E}\left[\|X_t - \tilde{X}_t\|_2^2\right] \leq (t+1)\left(R^2\beta + \alpha^2 + 4R_{\max}\sqrt{R^2\beta + \alpha^2}\right) + \frac{t}{k}\eta^2.$$

*Consequently,*

$$W_2^2(X_k, \tilde{X}_k) \leq (k+1)\left(R^2\beta + \alpha^2 + 4R_{\max}\sqrt{R^2\beta + \alpha^2}\right) + \eta^2.$$

*Proof.* First observe that, if $g$ is a standard Gaussian random vector in $\mathbb{R}^d$ and $\tilde{g}$ has the law of a standard Gaussian restricted to a ball of radius $\sqrt{d} \log(dk)$, then,

$$W_2^2(g, \tilde{g}) \leq \mathbb{P}\left(\|g\| > \sqrt{d} \log(dk)\right) \mathbb{E}\left[\|g\|^2\right] \leq \frac{1}{k}. \tag{27}$$

Indeed, if $\gamma$ and $\tilde{\gamma}$ are the respective laws of $g$ and $\tilde{g}$, there is a decomposition,

$$\gamma = \mathbb{P}\left(\|g\| \leq \sqrt{d} \log(dk)\right) \tilde{\gamma} + \mathbb{P}\left(\|g\| > \sqrt{d} \log(dk)\right) \gamma',$$

where $\gamma'$ is $\gamma$ conditioned on being outside the ball of radius $\sqrt{d}\log(dk)$. This decomposition induces a coupling between $g$ and $\tilde{g}$ which affords the bound in (27). The second inequality in (27) follows from $g$ being sub-Gaussian.

We now prove the claim by induction on $t$. The following observation, that arises from the definition of the noisy oracles, will be instrumental: One may decompose $\mathbb{E}\left[\|P_K(\tilde{X}_{t-1}+\eta\tilde{g}_{t-1})-\tilde{P}_K(\tilde{X}_{t-1}+\eta\tilde{g}_{t-1})\|_2^2\right]$ on the event $\{\|P_K(\tilde{X}_{t-1}+\eta\tilde{g}_{t-1})-\tilde{P}_K(\tilde{X}_{t-1}+\eta\tilde{g}_{t-1})\|_2 < \alpha\}$ to obtain,

$$\mathbb{E}\left[\|P_K(\tilde{X}_{t-1}+\eta\tilde{g}_{t-1})-\tilde{P}_K(\tilde{X}_{t-1}+\eta\tilde{g}_{t-1})\|_2^2\right] \le R^2\beta+\alpha^2. \tag{28}$$

The same argument also shows,

$$\mathbb{E}\left[\|X_0-\tilde{X}_0\|_2^2\right] = \mathbb{E}\left[\|S(K)-\tilde{S}(K)\|_2^2\right] \le R^2\beta+\alpha^2.$$

This establishes the base case of the induction, when $t=0$. For $t>0$, couple the processes $X_t$ and $\tilde{X}_t$, by coupling $g_{t-1}$ and $\tilde{g}_{t-1}$ according to the coupling in (27), and observe

$$\mathbb{E}\left[\|X_t-\tilde{X}_t\|_2^2\right] = \mathbb{E}\left[\|P_K(X_{t-1}+\eta g_{t-1})-\tilde{P}_K(\tilde{X}_{t-1}+\eta\tilde{g}_{t-1})\|_2^2\right]$$
$$= \mathbb{E}\left[\|P_K(X_{t-1}+\eta g_{t-1})-P_K(\tilde{X}_{t-1}+\eta\tilde{g}_{t-1})+P_K(\tilde{X}_{t-1}+\eta\tilde{g}_{t-1})-\tilde{P}_K(\tilde{X}_{t-1}+\eta\tilde{g}_{t-1})\|_2^2\right].$$

We also have, from (28), and with Cauchy-Schwartz,

$$\mathbb{E}\left[\langle P_K(X_{t-1}+\eta g_{t-1})-P_K(\tilde{X}_{t-1}+\eta\tilde{g}_{t-1}), P_K(\tilde{X}_{t-1}+\eta\tilde{g}_{t-1})-\tilde{P}_K(\tilde{X}_{t-1}+\eta\tilde{g}_{t-1})\rangle\right]$$
$$\le 2R_{\max}\mathbb{E}\left[\|P_K(\tilde{X}_{t-1}+\eta\tilde{g}_{t-1})-\tilde{P}_K(\tilde{X}_{t-1}+\eta\tilde{g}_{t-1})\|_2\right] \le 2R_{\max}\sqrt{R^2\beta+\alpha^2}$$

Combining the previous two calculations, we see,

$$\mathbb{E}\left[\|X_t-\tilde{X}_t\|_2^2\right] \le \mathbb{E}\left[\|P_K(X_{t-1}+\eta g_{t-1})-P_K(\tilde{X}_{t-1}+\eta\tilde{g}_{t-1})\|_2^2\right]$$
$$+ \mathbb{E}\left[\|P_K(\tilde{X}_{t-1}+\eta\tilde{g}_{t-1})-\tilde{P}_K(\tilde{X}_{t-1}+\eta\tilde{g}_{t-1})\|_2^2\right] + 4R_{\max}\sqrt{R^2\beta+\alpha^2}$$
$$\le R^2\beta+\alpha^2 + 4R_{\max}\sqrt{R^2\beta+\alpha^2}$$
$$+ \mathbb{E}\left[\|P_K(X_{t-1}+\eta g_{t-1})-P_K(\tilde{X}_{t-1}+\eta\tilde{g}_{t-1})\|_2^2\right]$$
$$\le R^2\beta+\alpha^2 + 4R_{\max}\sqrt{R^2\beta+\alpha^2}$$
$$+ \mathbb{E}\left[\|X_{t-1}+\eta g_{t-1}-(\tilde{X}_{t-1}+\eta\tilde{g}_{t-1})\|_2^2\right]$$
$$= R^2\beta+\alpha^2 + 4R_{\max}\sqrt{R^2\beta+\alpha^2}$$
$$+ \mathbb{E}\left[\|X_{t-1}-\tilde{X}_{t-1}\|_2^2\right] + \eta^2\mathbb{E}\left[\|g_{t-1}-\tilde{g}_{t-1}\|_2^2\right]$$
$$\le R^2\beta+\alpha^2 + 4R_{\max}\sqrt{R^2\beta+\alpha^2}$$
$$+ t\left(R^2\beta+\alpha^2 + 4R_{\max}\sqrt{R^2\beta+\alpha^2} + \frac{(t-1)\eta^2}{k}\right) + \frac{\eta^2}{k}$$
$$\le (t+1)\left(R^2\beta+\alpha^2 + 4R_{\max}\sqrt{R^2\beta+\alpha^2}\right) + \frac{t}{k}\eta^2.$$

The third inequality follows from the fact that, since $K$ is convex, $P_K$ is a contraction, and the penultimate inequality is the induction hypothesis, along with (27) and the independence of $g_{k-1}$ and $X_{k-1}$. $\qquad\square$

We now identify a regime for the noise parameters $(\tilde{\alpha},\beta,R)$ in which the dynamics in (26) have comparable guarantees to the ones in (25).

**Lemma 29.** *Suppose that $K$ contains a ball or radius $R_{\min}$, centered at $S(K)$, and is contained in a ball of radius $R_{\max}$. Let $\alpha \in (0,1)$, set $\eta = \tilde{\Theta}\left(\frac{R_{\min}^2}{(R_{\max}+1)^4}\frac{\alpha^2}{d}\right)$, $k = \tilde{\Theta}\left(\frac{(R_{\max}+1)^6}{R_{\min}^2}\frac{d}{\alpha^2}\right)$, and*

*assume that, for some $R > 0$, $\tilde{P}_K$ and $\tilde{S}(K)$ are $(\tilde{\alpha}, \beta, R)$-noisy oracles for $P_K$ and $S(k)$. Moreover assume*

$$R^2\beta + \tilde{\alpha}^2 + 4R_{\max}\sqrt{R^2\beta + \tilde{\alpha}^2} \leq \frac{d\alpha}{2k}.$$

*Then,*

$$\frac{1}{d}W_2^2(\tilde{X}_k, U_K) \leq 9\alpha,$$

*where $U_K$ is a random vector, uniformly distributed over $K$.*

*Proof.* We use the triangle inequality, followed by Theorem 26 and Lemma 28,

$$\frac{1}{\sqrt{d}}W_2(\tilde{X}_k, U_K) \leq \frac{1}{\sqrt{d}}W_2(\tilde{X}_k, X_k) + \frac{1}{\sqrt{d}}W_2(X_k, U_K)$$

$$\leq \sqrt{\frac{k+1}{d}\left(R^2\beta + \tilde{\alpha}^2 + 4R_{\max}\sqrt{R^2\beta + \tilde{\alpha}^2}\right) + \eta^2} + \sqrt{\alpha} \leq 3\sqrt{\alpha},$$

where we have also used that $\eta^2 \leq \alpha$. $\qquad\square$

### D.5 Towards a Private Sampling Algorithm: a Private Noisy Projection Oracle

Let $K \subset \mathbb{R}^d$ be a convex body and recall the projection operator, $P_K : \mathbb{R}^d \to \mathbb{R}^d$ given by,

$$P_K(x) = \arg\min_y\{y \in K | \|y - x\|_2\}.$$

We first show that when $X$ follows an admissible distribution, one can privately estimate $P_{F_q(X)}(x)$. To this end, we prove Lemma 14. The proof uses the following classical result, see [AW93, Proposition 5.3].

**Proposition 30.** *Fix $R > 0$ and let $K_1, K_2 \subset B(0, R)$ be two convex bodies. If $x \in \mathbb{R}^d$, then*

$$\|P_{K_1}(x) - P_{K_2}(x)\|_2 \leq 2\sqrt{(\|x\|_2 + R)}\delta_{\text{Haus}}(K_1, K_2)^{1/2}.$$

Combining the robustness of the projection operator with Lemma 8, we now prove Lemma 14.

*Proof of Lemma 14.* Similar to the proof of Lemma 12, we see that, under the assumption,

$$\delta_q(F_q(X), F_q(Y)) \leq \frac{R_{\min}}{4},$$

Lemma 8 implies,

$$\delta_{\text{Haus}}(F_q(X), F_q(Y)) \leq 6\frac{R_{\max} + r}{R_{\min}}\delta_q(F_q(X), F_q(Y)).$$

The above bound holds provided $F_q(X)$ contains a ball of radius $R_{\min}/2$, and when $F_q(X), F_q(Y)$ are contained in a ball of radius $R_{\max} + r$, centered at the origin. We invoke Proposition 30, according to which

$$\|P_{F_q(X)}(x) - P_{F_q(Y)}(x)\|_2 \leq 5\sqrt{(\|x\|_2 + R_{\max} + r)\frac{R_{\max} + r}{R_{\min}}}\delta_q(F_q(X), F_q(Y))^{1/2}$$

which completes the proof. $\qquad\square$

Having established that $P_{(.)}(x)$ is an approximate $\frac{1}{2}$-Hölder function with respect to the convex body, we prove the following corollary.

**Corollary 31.** *Let $\mathcal{D} \in A_q(R_{\max}, R_{\min}, r, L)$ be an admissible measure on $\mathbb{R}^d$. Then, for every $x \in \mathbb{R}^d$, there is an $\varepsilon$-differentially private algorithm $\mathcal{A}$ which for input $X = (X_1, \ldots, X_n)$ with i.i.d. entries from $\mathcal{D}$ satisfies for all $\alpha < \frac{\min\{r, R_{\min}\}}{2}$ that*

$$\mathbb{P}(\|\mathcal{A}(X) - P_{F_q(\mathcal{D})}(x)))\|_2 \leq \alpha) \geq 1 - \beta,$$

*for some*

$$n = \tilde{O}\left(\frac{(\|x\|_2 + R_{\max} + r)^4}{R_{\min}^2}\left(\frac{\left(d + \log\left(\frac{1}{\beta}\right)\right)}{\alpha^4 L^2} + \frac{d^3\log\left(\frac{1}{\beta}\right)^2 + d^2\log\left(\frac{1}{\beta}\right)^3}{\varepsilon^2\alpha^2 L} + \frac{d^3 + d^2\log(\frac{1}{\beta})}{\varepsilon^2\min\{r, R_{\min}\}^2 L}\right)\right).$$

*Proof.* By Lemma 14 $P_{(\cdot)}(x)$ is an approximate $\frac{1}{2}$-Hölder function with constant, $K = 5\sqrt{\frac{(\|x\|_2 + R_{\max} + r)(R_{\max} + r)}{R_{\min}}}$, with $h = \frac{1}{2}$ and $M = d$. $\qquad\square$

### D.6 A Private Sampling Algorithm for the Floating Body

We now focus on building the private sampling algorithm for the floating body, $F_q(\mathcal{D})$, of a distribution $\mathcal{D}$. Recall that we have access to i.i.d. samples drawn from $\mathcal{D}$ and, in light the previous section, we will use the i.i.d. samples to privately approximate the Steiner point and the projection operator for $F_q(\mathcal{D})$. To be more specific, notice first that the private algorithms defined in Corollaries 13 and 31 naturally produce noisy oracles for the Steiner point and the projection operators, respectively. The next Lemma exactly quantifies it in a convenient way for what follows.

**Lemma 32.** *Let $k \geq d$ and substitute $\tilde{\alpha} := \frac{d\alpha}{32k(R_{\max}+1)}$ for $\alpha$, and $\beta = \frac{d^2\alpha^2}{2^{14}k^2(R_{\max}+1)^2(R_{\max}+r)^2}$, in Corollary 31. Then, the algorithm promised by the Corollary is $\varepsilon$-differentially private and a $(\tilde{\alpha}, \beta, 4(R_{\max} + r))$ -noisy projection oracle for $F_q(\mathcal{D})$, which uses*

$$n = \tilde{O}\left(\frac{(\|x\|_2 + R_{\max} + r)^4}{R_{\min}^2}\left(\frac{(R_{\max}+1)^4 k^4}{d^3\alpha^4 L^2} + \frac{(R_{\max}+1)^2 k^2 d}{\varepsilon^2\alpha^2 L} + \frac{d^3}{\varepsilon^2 \min\{r, R_{\min}\}^2 L}\right)\right)$$

*samples.*

*Moreover, with the same parameters and the same bound on the number of samples, the output of the algorithm from Corollary 13 is a noisy oracle for $S(K)$.*

*Proof.* Fix $x \in \mathbb{R}^d$. If $\mathcal{A}(X)$ is the output of the algorithm in Corollary 31, we have

$$\mathbb{P}\left(\|P_{F_q(\mathcal{D})}(x) - \mathcal{A}(X)\|_2 \leq \frac{d}{32k(R_{\max}+1)}\alpha\right) \geq 1 - \frac{d^2\alpha^2}{2^{14}k^2(R_{\max}+1)^2(R_{\max}+r)^2}.$$

Moreover, by the construction in (20), almost surely,

$$\|P_{F_q(\mathcal{D})}(x) - \mathcal{A}(X)\|_2 \leq 4(R_{\max} + r).$$

The sample complexity bound follows by appropriately substituting terms.

The proof for the Steiner point is identical, with the bounds obtained in Corollary 13. $\qquad\square$

Now, recall that Lemma 29 reveals the level of the noise tolerance of the oracles under which the Langevin still produces an approximately uniform point of the floating body. Our final step is to combine Lemma 32 with Lemma 29 to complete the proof of Corollary 15.

*Putting it all together: Proof of Corollary 15.* Set $\eta = \tilde{\Theta}\left(\frac{R_{\min}^2}{(R_{\max}+1)^4}\frac{\alpha^2}{d}\right)$, $k = \tilde{\Theta}\left(\frac{(R_{\max}+1)^6}{R_{\min}^2}\frac{d}{\alpha^2}\right)$.
We shall invoke Lemma 32 to privately compute the initialization and each iteration of the noisy Langevin process, as in (26). For $(\tilde{\alpha}, \beta, R)$ as defined by Lemma 32, we have,

$$R^2\beta + \tilde{\alpha}^2 = 16(R_{\max}+r)^2\frac{d^2\alpha^2}{2^{14}k^2(R_{\max}+1)^2(R_{\max}+r)^2} + \frac{d^2\alpha^2}{1024k^2(R_{\max}+1)^2} = \frac{d^2\alpha^2}{512k^2(R_{\max}+1)^2}.$$

By our choice of $k$, we may freely assume $k \geq d$ and since $\alpha \leq 1$, we have, $\frac{d\alpha}{128k(R_{\max}+1)^2} \leq 1$. Thus,

$$R^2\beta + \tilde{\alpha}^2 + 4R_{\max}\sqrt{R^2\beta + \tilde{\alpha}^2} \leq \frac{d^2\alpha^2}{512k^2(R_{\max}+1)^2} + \frac{d\alpha}{4k} \leq \frac{d\alpha}{2k},$$

and Lemma 29 shows that, for the random vector $U_q$,

$$\frac{1}{d}W_2^2(\tilde{X}_k, U_q) \leq 9\alpha.$$

Moreover, note that by definition of the noisy projection oracle, we have, for every $t \leq k$,

$$\|\tilde{X}_t\|_2 \leq R_{\max} + 4(R_{\max} + r) \leq 5(R_{\max} + r).$$

Hence, recalling that $\tilde{g}_t$ has the law of a standard Gaussian conditioned on the ball of radius $\sqrt{d}\log(dk)$, we have, since $R_{\max} \geq R_{\min}$,

$$\|\tilde{X}_t + \eta\tilde{g}_t\|_2 \leq 5(R_{\max} + r) + \eta\sqrt{d}\log(dk) \leq \tilde{O}\left(R_{\max} + r + \log(k)\frac{R_{\min}^2}{(R_{\max} + 1)^4}\right)$$

$$\leq \tilde{O}\left(R_{\max} + r + \log\left(\frac{d}{\alpha}\right)\frac{R_{\min}^2}{(R_{\max} + 1)^4}\right) = \tilde{O}\left(R_{\max} + r + 1\right).$$

Thus, since Lemma 32 is invoked $k + 1$ times, each time with an $x$ satisfying $\|x\|_2 \leq \tilde{O}\left(R_{\max} + r + 1\right)$, the sample complexity is

$$n = \tilde{O}\left(\frac{(R_{\max} + r + 1)^8}{R_{\min}^2}\left(\frac{k^5}{d^3\alpha^4 L^2} + \frac{k^3 d}{\varepsilon^2\alpha^2 L} + \frac{kd^3}{\varepsilon^2\min\{r, R_{\min}\}^2 L}\right)\right).$$

Substituting $k$, we get

$$n = \tilde{O}\left(\frac{(R_{\max} + r + 1)^{38}}{R_{\min}^{12}}\left(\frac{d^2}{\alpha^{14}L^2} + \frac{d^4}{\varepsilon^2\alpha^8 L} + \frac{d^4}{\varepsilon^2\alpha^2\min\{r, R_{\min}\}^2 L}\right)\right).$$

$\square$

## D.7 Beyond Uniform Sampling

Let us note that Theorem 26 is a specialized form of a more general result. In fact, [Leh21, Theorem 2] offers sampling guarantees for so-called log-concave measures, that is measures with densities of the form

$$e^{-\varphi(x)}\mathbf{1}_K(x),$$

where $K$ is a convex body, and $\varphi(x)$ is a convex function (the uniform sampling simply sets $\phi$ to be constant). A straightforward adaption of our differentially private projection oracle can lead to a differentially private sampler from arbitrary log-concave measures supported on the floating body. The sample complexity would now need to also depend, polynomially, on the Lipschitz constant of $\varphi$; we leave the exact dependence on it for future work.

We chose to state and prove Corollary 5 for the uniform measure on $F_q(\mathcal{D})$. This decision was made both for the sake of simplicity, but also because, arguably, the uniform measure is among the most interesting cases; one may need to "exclude outliers" and a uniform sample produces a typical representative from what remains.

However, let us note one possible application of (low-temperature) sampling from more general measures, which may be of interest in future research in the differential privacy community. It is well known that when $\varphi$ is a convex function, sampling from the measure $e^{-\varphi(x)}\mathbf{1}_K(x)$ is intimately connected to *optimization*(as in [RRT17]). That is, sampling is connected to finding

$$\arg\max_{x \in K}\varphi(x).$$

Thus, when considering floating bodies, one should be able to privately optimize convex functions over $F_q(\mathcal{D})$, which can be seen as a given data-set "pruned" to have no outliers.

## E    Examples of Admissible Distributions

In this section we demonstrate one prototypical example of a class of admissible distributions. The example should serve to both show that Definition 2 is not vacuous as well as give the reader some idea of the possible interplay between the different parameters in the definition. We focus on log-concave measures but mention that similar reasoning can be applied to other classes, like $\alpha$-stable laws, see [AR20, Section 7].

**Symmetric log-concave distributions**    A measure $\mathcal{D}$ on $\mathbb{R}^d$ is said to be log-concave if its has a density $e^{-\varphi(x)}$, such that $\varphi$ is a convex function. Prominent examples of log-concave measures include Gaussians and uniform measures on convex sets. To enforce a useful normalization we shall consider isotropic measures. These are measures whose expectation is zero, and whose covariance matrix equals the identity. Finally, we say that a distribution is symmetric, if when $X \sim \mathcal{D}$, then $X$ and $-X$ have the same law. Our main result for log-concave measures is as follows.

**Proposition 33.** *Let $\mathcal{D}$ be a symmetric, log-concave, and isotropic measure on $\mathbb{R}^d$, and let $q \in (\frac{1}{2}, 1)$. Then, $\mathcal{D} \in A_q \left( q - \frac{1}{2}, \log\left(\frac{1}{2(1-q)}\right), \frac{1-q}{2}, \frac{1-q}{8} \right)$. That is, one can take,*

$$R_{\min} = q - \frac{1}{2}, R_{\max} = \log\left(\frac{1}{2(1-q)}\right), r = \frac{1-q}{2}, L = \frac{1-q}{8}.$$

The proof of Proposition 33 is broken down in several lemmas. We begin by showing that in every direction, the $q$ quantile has some mass around it.

**Lemma 34.** *Fix any $q \in (\frac{1}{2}, 1)$, and $X \sim \mathcal{D}$, a symmetric, log-concave, and isotropic measure on $\mathbb{R}^d$. Then, if $L = \frac{(1-q)^2}{16}, r = \frac{(1-q)}{2}$, $X$ satisfies that for any direction $\theta \in \mathbb{S}^{d-1}$, it holds, for the density $f_\theta$ of $\mathcal{D}_\theta := \langle X, \theta \rangle$, that*

$$f_\theta(t) > L$$

*when*

$$t \in [Q_q(\mathcal{D}_\theta) - r, Q_q(\mathcal{D}_\theta) + r].$$

*Proof.* For $\theta \in \mathbb{S}^{d-1}$, by the Prékopa-Leindler inequality, $\mathcal{D}_\theta$ is a symmetric, log-concave, and isotropic measure on $\mathbb{R}$ (see [LV07, Theorem 5.1]). So, by [AR20, Lemma 13], $f_\theta(Q_q(\mathcal{D}_\theta)) \geq \frac{1-q}{4}$. Since $f_\theta$ is symmetric and uni-modal it is decreasing on $(0, \infty)$. So, for any $t < Q_q(\mathcal{D}_\theta)$, $f_\theta(t) \geq \frac{1-q}{4}$, as well. $t > Q_q(\mathcal{D}_\theta)$ note that if we take $r < \frac{1-q}{2}$ then, since $f_\theta \leq 1$, as in [LV07, Lemma 5.5],

$$\int_{-\infty}^{m_q(\mathcal{D}_\theta)+r} f_\theta(t)dt = q + \int_{Q_q(\mathcal{D}_\theta)}^{Q_q(\mathcal{D}_\theta)+r} f_\theta(t)dt \leq q + \int_{Q_q(\mathcal{D}_\theta)}^{Q_q(\mathcal{D}_\theta)+\frac{1-q}{2}} dt \leq \frac{q+1}{2}.$$

Hence, $Q_q(\mathcal{D}_\theta) + r \leq Q_{\frac{q+1}{2}}(\mathcal{D}_\theta)$ and, the same argument as before shows, $f_\theta(Q_q(\mathcal{D}_\theta) + r) \geq f_\theta(m_{\frac{q+1}{2}}(\mathcal{D}_\theta)) \geq \frac{1-q}{8}$. In particular, this is true for any $t \in [Q_q(\mathcal{D}_\theta), Q_q(\mathcal{D}_\theta) + r]$. $\square$

The fact that for log-concave measure the floating body contains a ball, and is contained in a ball of fixed radii was previously proven in [AR20].

**Lemma 35** ([AR20, Lemma 13]). *Fix any $q \in (\frac{1}{2}, 1)$, and $X \sim \mathcal{D}$, a symmetric, log-concave, and isotropic measure on $\mathbb{R}^d$. Then, for every $\theta \in \mathbb{S}^{d-1}$,*

$$q - \frac{1}{2} \leq Q_q(\langle X, \theta \rangle) \leq 1 + \log\left(\frac{1}{2(1-q)}\right).$$

Let us now prove Proposition 33.

*Proof of Proposition 33.* In light of Lemma 34 and Lemma 35 it will be enough to show that if $\{X_i\}_{i=1}^n$ are *i.i.d.* as $\mathcal{D}$, then,

$$\mathbb{P}\left(\max_i \|X_i\|_2 > 10\sqrt{d}n^3\right) \leq e^{-n}.$$

But this is clear, since each $X_i$ has sub-exponential tails, [LV07, Lemma 5.17], and $\mathbb{E}[\|X_i\|_2^2] = d$. Indeed, with a union-bound,

$$\mathbb{P}\left(\max_i \|X_i\|_2 > 10\sqrt{d}n^3\right) \leq n\mathbb{P}\left(\|X_1\|_2 > 10\sqrt{d}n^3\right) \leq ne^{-10n^2+1} \leq e^{-n}.$$

$\square$

# F   On the Extension Lemma and the Proof of Theorem 10

In this section we provide for the interested reader more details on the Extension Lemma (stated in Proposition 6) and how it is used in Section B.4 to construct the final private algorithm to establish Theorem 10. We repeat here the Extension Lemma for convenience.

**Proposition 36** ("The Extension Lemma" Proposition 2.1, [BCSZ18a] ). *Let $\hat{\mathcal{A}}$ be an $\varepsilon$-differentially private algorithm designed for input from $\mathcal{H} \subseteq (\mathbb{R}^d)^n$ with arbitrary output measure space $(\Omega, \mathcal{F})$. Then there exists a randomized algorithm $\mathcal{A}$ defined on the whole input space $(\mathbb{R}^d)^n$ with the same output space which is $2\varepsilon$-differentially private and satisfies that for every $X \in \mathcal{H}$, $\mathcal{A}(X) \stackrel{d}{=} \hat{\mathcal{A}}(X)$.*

We start with recalling that the input space is $\mathcal{M} = (\mathbb{R}^d)^n$ equipped with the Hamming distance and the output space is $(\mathbb{R}^M, \|\cdot\|_p)$ equipped with the Lebesgue measure. Furthermore let us assume also that for any $X \in \mathcal{H} = \mathcal{H}(A)$ the randomised restricted algorithm $\hat{\mathcal{A}}(X)$ follows a $\mathbb{R}^M$-valued continuous distribution with a density $f_{\hat{\mathcal{A}}(X)}$ with respect to the Lebesgue measure, given by (20), (21). Applying the Extension Lemma readily gives the $\varepsilon$-differentially private extension $\mathcal{A}$ on input $X \in (\mathbb{R}^d)^n$ which agrees with $\hat{\mathcal{A}}$ when $X \in \mathcal{H}$.

**Density of the extended algorithm**   Now one may wonder if the extended algorithm admits a density and, if so, what does it look like. It turns out that because the restricted algorithm admits a density with respect to the Lebesgue measure, the same holds for the extended algorithm. Moreover, the density has, in fact, a simple-to-state formula. In more detail, an inspection of the proof of the Extension Lemma [Section 4, [BCSZ18a]] gives that the "extended" $\varepsilon$-differentially private algorithm $\mathcal{A}$ admits a density, on input $X \in (\mathbb{R}^d)^n$, given by

$$f_{\mathcal{A}(X)}(\omega) = \frac{1}{Z_X} \inf_{X' \in \mathcal{H}} \left[ \exp\left(\frac{\varepsilon}{4} d_H(X, X')\right) f_{\hat{\mathcal{A}}(X')}(\omega) \right], \omega \in \mathbb{R}^M \tag{29}$$

where

$$Z_X := \int_{\mathbb{R}^M} \inf_{X' \in \mathcal{H}} \left[ \left(\frac{\varepsilon}{4} d_H(X, X')\right) f_{\hat{\mathcal{A}}(X)'}(\omega) \right] d\omega.$$

For reasons of completeness we state here (and prove in the following section) the corollary of the Extension Lemma that establishes that in our setting the algorithm $\mathcal{A}$ satisfies the desired properties of the Extension Lemma.

**Proposition 37.** *Under the above assumptions, the algorithm $\mathcal{A}$ with density given in (29) is $\varepsilon$-differentially private and for every $X' \in \mathcal{H}$, $\mathcal{A}(X') \stackrel{d}{=} \hat{\mathcal{A}}(X')$.*

As a technical remark note that, in order for the density in equation (29) to be well-defined we require that, for every $X \in (\mathbb{R}^d)^n$ the "unnormalized" density function

$$G_X(\omega) := \inf_{X' \in \mathcal{H}} \left[ \exp\left(\frac{\varepsilon}{2} d_H(X, X')\right) f_{\hat{\mathcal{A}}(X')}(\omega) \right], \omega \in \mathbb{R}^M \tag{30}$$

is integrable. This condition follows from the following Lemma, which establishes - among other properties - that $G_X$ is a continuous function almost everywhere and has a finite integral.

**Lemma 38.** *Suppose the above assumptions hold and fix $X \in \mathcal{H}(A)$. Then,*

- *$G_X(\omega) = 0$ for all $\omega \notin \mathcal{I} = \{\|\omega\|_p \leq 2K(R + r/2)\}$.*

- *$G_X$ is $\mathcal{R}$-Lipschitz on $\mathcal{I}$ with a universal (that is, independent of the value of $X \in \mathcal{H}$) Lipschitz constant $\mathcal{R} < \infty$.*

*Furthermore, it holds $0 \leq \int_{\omega \in \mathbb{R}} G_X(\omega) d\omega \leq 1$.*

Finally, notice that the infimum over $X' \in \mathcal{H}$ in (29) is the main reason the extension lemma comes with no explicit termination time guarantees; the termination time largely depends on the "nature" of $\mathcal{H}$. See the appendix of [TVGZ20] for a further discussion on this point.

### F.1   Proof of Proposition 37

We start with using [BCSZ18a, Lemma 4.1.] applied to our setting. This gives the following Lemma.

**Lemma 39.** *Let $\mathcal{A}'$ be a real-valued randomized algorithm designed for input from $\mathcal{H}' \subseteq (\mathbb{R}^d)^n$. Suppose that for any $X \in \mathcal{H}'$, $\mathcal{A}'(X)$ admits a density function with respect to the Lebesgue measure $f_{\mathcal{A}'(X)}$. Then the following are equivalent*

- *(1) $\mathcal{A}'$ is $\varepsilon$-differentially private on $\mathcal{H}$;*

*(2) For any $X, X' \in \mathcal{H}$*

$$f_{\mathcal{A}'(X)}(\omega) \leq e^{\varepsilon d_H(X,X')} f_{\mathcal{A}'(X')}(\omega), \tag{31}$$

*almost surely with respect to the Lebesgue measure.*

*Proof of Proposition 37.* We first prove that $\mathcal{A}$ is $\varepsilon$-differentially private over all pairs of input from $(\mathbb{R}^d)^n$. Using Lemma 39 it suffices to prove that for any $X_1, X_2 \in \mathcal{H}$,

$$f_{\mathcal{A}(X_1)}(\omega) \leq \exp\left(\varepsilon d_H(X_1, X_2)\right) f_{\mathcal{A}(X_2)}(\omega),$$

almost surely with respect to the Lebesgue measure. We establish it in particular for every $\omega \in \mathbb{R}^M$. Notice that if $\omega \notin \mathcal{I}$, both sides are zero from Lemma 38. Hence let us assume $\omega \in \mathcal{I}$. Let $X_1, X_2 \in \mathbb{R}^n$. Using triangle inequality we obtain for every $\omega \in \mathcal{I}$,

$$\inf_{X' \in \mathcal{H}} \left[\exp\left(\frac{\varepsilon}{2} d_H(X_1, X')\right) f_{\hat{\mathcal{A}}(X')}(\omega)\right] \leq \inf_{X' \in \mathcal{H}} \left[\exp\left(\frac{\varepsilon}{2}\left[d_H(X_1, X_2) + d_H(X_2, X')\right]\right) f_{\hat{\mathcal{A}}(X')}(\omega)\right]$$

$$= \exp\left(\frac{\varepsilon}{2} d_H(X_1, X_2)\right) \inf_{X' \in \mathcal{H}} \left[\exp\left(\frac{\varepsilon}{2} d_H(X, X')\right) f_{\hat{\mathcal{A}}(X')}(\omega)\right],$$

which implies that for any $X_1, X_2 \in \mathcal{M}$,

$$Z_{X_1} = \int_{\Omega} \inf_{X' \in \mathcal{H}} \left[\exp\left(\frac{\varepsilon}{2} d(X_1, X')\right) f_{\hat{\mathcal{A}}(X')}(\omega)\right] d\omega$$

$$\leq \exp\left(\frac{\varepsilon}{2} d(X_1, X_2)\right) \int_{\Omega} \inf_{X' \in \mathcal{H}} \left[\exp\left(\frac{\varepsilon}{2} d(X_2, X')\right) f_{\hat{\mathcal{A}}(X')}(\omega)\right] d\omega$$

$$= \exp\left(\frac{\varepsilon}{2} d(X_1, X_2)\right) Z_{X_2}.$$

Therefore using the above two inequalities we obtain that for any $X_1, X_2 \in \mathbb{R}^n$ and $\omega \in \mathcal{I}$,

$$f_{\mathcal{A}(X_1)}(\omega) = \frac{1}{Z_{X_1}} \inf_{X' \in \mathcal{H}} \left[\exp\left(\frac{\varepsilon}{2} d_H(X_1, X')\right) f_{\hat{\mathcal{A}}(X')}(\omega)\right]$$

$$\leq \frac{1}{\exp\left(-\frac{\varepsilon}{2} d_H(X_2, X_1)\right) Z_{X_2}} \exp\left(\frac{\varepsilon}{2} d_H(X_1, X_2)\right) \inf_{X' \in \mathcal{H}} \left[\exp\left(\frac{\varepsilon}{2} d(X_2, X')\right) f_{\hat{\mathcal{A}}(X')}(\omega)\right]$$

$$= \exp\left(\frac{\varepsilon}{2} d(X_1, X_2)\right) \frac{1}{Z_{X_2}} \inf_{X' \in \mathcal{H}} \left[\exp\left(\frac{\varepsilon}{2} d_H(X_2, X')\right) f_{\hat{\mathcal{A}}(X')}(\omega)\right]$$

$$= \exp\left(\varepsilon d_H(X_1, X_2)\right) f_{\mathcal{A}(X_2)}(\omega),$$

as we wanted.

Now we prove that for every $X \in \mathcal{H}$, $\mathcal{A}(X) \stackrel{d}{=} \hat{\mathcal{A}}(X)$. Consider an arbitrary $X \in \mathcal{H}$. We know that $\hat{\mathcal{A}}$ is $\varepsilon/2$-differentially private which based on Lemma 39 implies that for any $X, X' \in \mathcal{H}$

$$f_{\hat{\mathcal{A}}(X)}(\omega) \leq \exp\left(\frac{\varepsilon}{2} d_H(X, X')\right) f_{\hat{\mathcal{A}}(X')}(\omega), \tag{32}$$

almost surely with respect to the Lebesgue measure. Observing that the above inequality holds almost surely as equality if $X' = X$ we obtain that for any $X \in \mathcal{H}$ it holds

$$f_{\hat{\mathcal{A}}(X)}(\omega) = \inf_{X' \in \mathcal{H}} \left[\exp\left(\frac{\varepsilon}{2} d_H(X, X')\right) f_{\hat{\mathcal{A}}(X')}(\omega)\right],$$

almost surely with respect to the Lebesgue measure. Using that $f_{\hat{\mathcal{A}}(X)}$ is a probability density function we conclude that in this case

$$Z_X = \int f_{\hat{\mathcal{A}}(X)}(\omega) d\omega = 1.$$

Therefore

$$f_{\hat{\mathcal{A}}(X)}(\omega) = \frac{1}{Z_X} \inf_{X' \in \mathcal{H}} \left[\exp\left(\varepsilon d_H(X, X')\right) f_{\hat{\mathcal{A}}(X')}(\omega)\right],$$

almost surely with respect to the Lebesgue measure and hence

$$f_{\hat{\mathcal{A}}(X)}(\omega) = f_{\mathcal{A}(X)}(\omega),$$

almost surely with respect to the Lebesgue measure. This suffices to conclude that $\hat{\mathcal{A}}(X) \stackrel{d}{=} \mathcal{A}(X)$ as needed.

The proof of Proposition 37 is complete. $\qquad\square$

### F.2 Proof of Lemma 38

*Proof of Lemma 38.* First notice that if $\omega \notin \mathcal{I}$, from (20) for any $X' \in \mathcal{H}$, $f_{\hat{\mathcal{A}}(X')}(\omega) = 0$. Therefore indeed

$$0 \leq f_{\mathcal{A}(X)}(\omega) \leq \exp\left(\frac{\varepsilon}{2} d_H(X, X')\right) f_{\hat{\mathcal{A}}(X')}(\omega) = 0.$$

We prove now that for all $X' \in \mathcal{H}$, the function $\exp\left(\frac{\varepsilon}{2} d_H(X, X')\right) f_{\hat{\mathcal{A}}(X')}(\omega)$ is $\mathcal{R}$-Lipschitz on $\mathcal{I}$. The claim then follows by the elementary real analysis fact that the pointwise infimum over an arbitrary family of $\mathcal{R}$-Lipschitz functions is an $\mathcal{R}$-Lipschitz function.

Now recall that for all $a, b > 0$ by elementary calculus, $|e^{-a} - e^{-b}| \leq |a - b|$. Hence, for fixed $X' \in \mathcal{H}$, using the definition of the density in equation (20), we have for any $\omega, \omega' \in \mathcal{I}$,

$$|f_{\hat{\mathcal{A}}(X')}(\omega) - f_{\hat{\mathcal{A}}(X')}(\omega')|$$

$$\leq \frac{\varepsilon\left(\frac{Ln}{2W}\right)^h}{4\hat{Z}_{X'}} |\min\left\{K^{-1}\|\omega - f(F_q(X'))\|_p, \min\{r, R_{\min}\}\right\} - \min\left\{K^{-1}\|\omega' - f(F_q(X'))\|_p, \min\{r, R_{\min}\}\right\}|$$

Now combining with triangle inequality we conclude

$$|f_{\hat{\mathcal{A}}(X')}(\omega) - f_{\hat{\mathcal{A}}(X')}(\omega')| \leq \frac{\varepsilon\left(\frac{Ln}{2W}\right)^h}{4\hat{Z}_{X'}} \min\left\{K^{-1}\|\omega - \omega'\|_p, \min\{r, R_{\min}\}\right\} \leq \frac{\varepsilon\left(\frac{Ln}{2W}\right)^h}{4K\hat{Z}_{X'}}\|\omega - \omega'\|_p.$$

Now following the proof of the accuracy guarantee (Lemma 24) we have for all $X' \in \mathcal{H}$,

$$\hat{Z}_{X'} \geq \hat{Z} := \omega_{M,p}\left(\frac{4W^h}{(nL)^h \varepsilon}\right)^{M-1} \frac{(M-1)!}{2}. \tag{33}$$

Hence it holds

$$|f_{\hat{\mathcal{A}}(X')}(\omega) - f_{\hat{\mathcal{A}}(X')}(\omega')| \leq \frac{\varepsilon\left(\frac{Ln}{2W}\right)^h}{4K\hat{Z}}\|\omega - \omega'\|_p.$$

Finally, $\exp\left(\frac{\varepsilon}{2} d_H(X, X')\right)$ is a constant independent of $\omega$ with $\exp\left(\frac{\varepsilon}{2} d_H(X, X')\right) \leq \exp(\frac{\varepsilon n}{2})$.

Combining the above, $\exp\left(\frac{\varepsilon}{2} d_H(X, X')\right) f_{\hat{\mathcal{A}}(X')}(\omega)$ is $\mathcal{R} = \exp(\frac{\varepsilon n}{2})\frac{\varepsilon\left(\frac{Ln}{2W}\right)^h}{4K\hat{Z}}$-Lipschitz and notice that $\mathcal{R}$ is independent of $X'$. The proof of the Lipschitz continuity is complete.

The final part follows from the fact that $G$ is non-negative by definition. Moreover, again by definition, for arbitrary fixed $X' \in \mathcal{H}$, $f_{\hat{\mathcal{A}}(X')}$ integrates to one and upper bounds, pointwise, the function $G_X$. $\qquad \square$