# OpenReview forum: "Archimedes Meets Privacy: On Privately Estimating Quantiles in High Dimensions Under Minimal Assumptions"
_NeurIPS.cc/2022/Conference — NeurIPS 2022 Accept_

### Official Review · Reviewer_49vS · 2022-07-06

**Rating:** 6
**Confidence:** 3
**Soundness:** 3 good
**Presentation:** 3 good
**Contribution:** 3 good

**Summary:**

The paper makes several contributions to the theory of private estimation from high-dimensional distributions, with a main focus on a) quantile estimation for $M$ marginals (where $M$ can be larger than the dimension $d$) and b) privately reporting an interior point in the so-called "floating body" of the distribution.
Past work either makes relatively strong assumptions on the underlying distribution (e.g. subgaussian) or deal with worst-case distributions resulting in substantially worse sampling complexity. In this paper the authors seek a middle ground, identifying "minimal" sufficient assumptions under which $\varepsilon$-DP sampling bounds can be shown.

**Questions:**

Questions:
- Can you elaborate on the sense in which you are making "minimal assumptions"?
- Can you present, at least theoretically, a class of distributions on which your estimators outperform existing ones?

Smaller questions (no need to address in rebuttal):
- Please clarify two things concerning Definition 2: Can $c$ depend on $\theta$? Can poly$(n,d)$ depend on $\theta$?
- Line 91-93: I don't get this. Can't the same sample work with high probability for all $M$ directions, giving logarithmic dependence on $M$?
- Line 106: Are you assuming $M\geq d$ and ignoring polylogarithmic factors?

Comments:
- Lines 141: Typos
- Line 195: Missing last part of sentence

**Limitations:**

As far as I can tell the authors have done a good job at discussing limitations

**Strengths And Weaknesses:**

Strengths:
- The paper addresses an important type of questions in private statistics
- The technical contribution over past work appears solid (based on the authors' own description, I am not familiar with the past work)
- The simplified versions of the main results (Theorems 3 and 4) are appealing
- Uses tools from convex geometry (Steiner point of the floating body) that are novel in this context and may be useful more generally in private estimation
- The paper is well-written and clear

Weaknesses:
- The distributional assumptions made may be natural (given enough thought), but it is not clear in what sense they are "minimal" or "needed"
- For problem b) the improvement over worst-case distributions lies only in replacing approximate DP with pure DP, which is arguably mainly of theoretical interest
- The bound is on sample complexity only, there is no efficient algorithm; in particular, there is no empirical evidence that the new estimators can outperform existing ones in practice

---

> ### Author Response · Authors · 2022-08-02
> **Rebuttal for Reviewer 49vS**
>
> Thank you for taking the time to read our work and for providing thoughtful and helpful feedback. We are happy that you found the paper well-written and that you've appreciated our results and techniques.
>
> We hope that our rebuttal can address your concerns.
>
> - "Can you elaborate on the sense in which you are making "minimal assumptions"?"
>
> We understand the concern of whether our distributional assumptions are "needed" or "minimal", and agree that it would be important to elaborate more on the role of these assumptions. See below for the main details; We will add the resulting discussion to the next version of the paper.
>
> We believe we can safely characterize the assumptions from Definition 2 as minimal for the following reasons. First, note that assumptions (1) and (3) are indeed necessary for private quantile estimation. This is an outcome of the tight lower bound for the one-dimensional case (see the main result of [TVGZ20]), where it is established that the minimal sample complexity to privately learn one quantile of a distribution explodes to infinity whenever there is no bound on the possible quantile values ($R=R_{\max}=\infty$) or the distribution has no mass around the quantile ($L=0$). We highlight that the latter (having a positive density around the quantile) is a requirement for optimal estimation rates (and a standard assumption) for even non-private one-dimensional quantile estimation (e.g., see [1, Chs. 21, 25.3]). Now, assumption (2) is equivalent to the assumption that the floating body contains a ball of radius $R_{\min}$. Again, if $R_{\min}=0$, the floating body has an empty interior, and the floating body's estimation becomes a degenerate task. Finally, assumption (4) is a very mild boundedness technical condition that our data is arbitrarily polynomially-bounded, which almost does not hurt generality. For example, for the "heavy-tailed" Cauchy random vectors we can take the polynomial $dn$.
>
> - "For problem b) the improvement over worst-case distributions lies only in replacing approximate DP with pure DP, which is arguably mainly of theoretical interest"
>
> Indeed, in this context, the improvement over the worst case setting is of a theoretical nature. However, we emphasize that this goes far beyond replacing approximate DP with pure DP. The results concerning the worst case setting assume (the necessary condition in that context) that the sample lies **on a grid**. This is a very restrictive assumption, which we avoid by making distributional assumptions. Thus, not only can we improve from approximate to pure DP, but we can also address configurations that are intractable for previous methods.
>
> We also point out that the best known sample complexity bounds for the interior point problem in the worst case regime are quite different between the pure and approximate DP settings [BMNS'19, KSS'20]. For the pure DP case, the best known upper and lower bounds are $\tilde{O}(d^4)$ and $\tilde{\Omega}(d^2)$ respectively, whereas for the approximate DP case, the corresponding bounds are $\tilde{O}(d^{2.5})$ and $\tilde{\Omega}(d)$.
>
> - "Can you present, at least theoretically, a class of distributions on which your estimators outperform existing ones?"
>
> In the appendix of our paper (Section E), we give the important example of log-concave and symmetric distributions as a class of distributions captured by our results. In light of our previous point, to the best of our knowledge, no existing estimators can handle this class in full generality.
> Other classes that satisfy our assumptions and for which we can derive meaningful bounds are that of $\alpha$-stable distributions (see [Definition 5, AR20]). We chose not to focus on this class but can definitely add the necessary details to the final version if you think it's a good idea.
>
> - "The bound is on sample complexity only, there is no efficient algorithm; in particular, there is no empirical evidence that the new estimators can outperform existing ones in practice"
>
> We agree that time-efficiency is one limitation of our work (and in fact of all prior work on the questions studied -- none of these run in polynomial time, except for special cases such as subgaussian distributions). The main bottleneck in our approach towards creating a time-efficient algorithm is the extension lemma, which requires solving a highly non-trivial optimization problem (see (Equation (27) in Section F). Proving time complexity bounds for our algorithms is a very interesting question, and we hope to address such problems in future works.
>
> [1] A. W. van der Vaart. Asymptotic Statistics. Cambridge Series in Statistical and Probabilistic Mathematics. Cambridge University Press, 1998.

---

> > ### Comment · Reviewer_49vS · 2022-08-08
> > **Comment on author rebuttal**
> >
> > Thanks for the thorough response which addresses my most important concerns. I will increase my score.

---

> > > ### Author Response · Authors · 2022-08-08
> > > **Thanks for response**
> > >
> > > We are thankful for your response and revised score. We would be happy of course to elaborate more if you have any further questions or concerns.

---

### Official Review · Reviewer_8PiM · 2022-07-09

**Rating:** 7
**Confidence:** 3
**Soundness:** 4 excellent
**Presentation:** 3 good
**Contribution:** 4 excellent

**Summary:**

This paper is on estimation of multivariate and high-dimensional quantiles under differential privacy constraints. Authors have considered the floating body representation of multivariate quantiles, which is related to Tukey depth. There are two streams of papers on differential privacy: one that considers worst-case scenario and the other that uses strong probabilistic assumptions. Unlike these mainstream lines of work, in this paper authors have attempted to use minimal probabilistic assumptions. Authors also propose privacy preserving interior point estimation and privacy preserving sampling from the floating body.


**Questions:**

Is it possible for the authors to discuss the technical assumptions more carefully and thoroughly, perhaps in the supplementary materials? Also provide a few examples of distributions and quantiles that satisfy these assumptions.


**Limitations:**

This is a theoretical work. In practice, running computations for all possible vectors on the unit ball $S_{d -1}$ is an impossible choice. Hence there has to be some more modifications when this work extends to algorithms and practical implementations.



**Strengths And Weaknesses:**

Strengths:


This is a study that has several interesting directions. The problem is important, and the paper and the supplementary materials are clearly written. While I did not check the technical details carefully, the arguments seem very transparent and clear in the portions of the proof of Theorem 7 that I glanced over. I think authors focus on moving away from both the stringent worst case scenario and also the strong assumption scenario is much welcome, and this is a significant contribution from that perspective. The comparison with related works from the literature shows both authors' understanding of the challenges and the areas that need further development.


Weaknesses:

While the technical assumptions are clearly stated, I could not find any discussion on them. For the assumptions in lines 71-77, does $R_{max}, R_{min}$ etc depend on $\theta$ or are these constants valid uniformly for al values of $\theta$? Depending on how it is interpreted, Assumption-2 (line 74) of Definition-2 in particular  seems a bit strange, and might be vacuously true.


There are quite a few typos, in particular with capitalization (or lack of capitalization) of proper nouns.

Also, I could not find any discussions on the limitations of this work, but since this is a purely theoretical work, this is less important.
In practice, running computations for all possible vectors on the unit ball $S_{d -1}$ is not a viable alternative. Hence there has to be some more modifications when this work extends to algorithms and practical implementations.

---

> ### Author Response · Authors · 2022-08-02
> **Rebuttal for Reviewer 8PiM**
>
> Thank you for the positive review and extensive feedback. These types of comments are extremely helpful, and we will implement the necessary changes in the next version.
>
> Regarding your specific comments/questions:
>
> - "While the technical assumptions are clearly stated, I could not find any discussion on them. For the assumptions in lines 71-77, does $R_{\max}, R_{\min}$ etc depend on $\theta$ or are these constants valid uniformly for all values of $\theta$ ? Depending on how it is interpreted, Assumption-2 (line 74) of Definition-2 in particular seems a bit strange, and might be vacuously true."
>
> All assumptions in Definition 2 should hold uniformly over the sphere, and should not depend on $\theta$. So, for example, the second item says that all one-dimensional quantiles ($Q_q(\langle X, \theta \rangle)$ for all $\theta$) lie on the interval $[-R_{\max}, R_{\max}]$ where $R_{\max}$ is independent of any specific $\theta$. Thank you for spotting this possible ambiguity in the definition, we will make sure to rewrite the definition so as to remove any ambiguity.
>
> - "There are quite a few typos, in particular with capitalization (or lack of capitalization) of proper nouns."
>
> We apologize for these typos. We are running independent passes for fixing these typos and making minor tweaks to the style.
>
> - "In practice, running computations for all possible vectors on the unit ball is not a viable alternative. Hence there has to be some more modifications when this work extends to algorithms and practical implementations."
>
> Indeed, our algorithm is mostly theoretical and we did not focus on producing an efficient implementation. We do wish to remark the following on this matter. For many computations inside the suggested algorithms (e.g., when computing the Steiner point of the floating body) there is no need to run computations for all possible vectors on the unit ball. The empirical floating body of a sample is always a polytope; i.e., in convex-geometry terms its support function is piece-wise constant, so (for example) the Steiner point of the floating body is always exactly computable in finite time. The main bottleneck for an algorithm is applying the  extension lemma, which requires solving a highly non-trivial optimization problem (see (Equation (27) in Section F) over the typical set. Solving these optimization problems with a finite (or ideally, polynomial) bound on the running time would require some clever discretization, potentially similar to the one proposed in the easier one-dimensional case in [TVGZ20]. Proving time complexity bounds for our algorithms is a very interesting question, and we hope to address this in future works.
>
> - "Is it possible for the authors to discuss the technical assumptions more carefully and thoroughly, perhaps in the supplementary materials? Also provide a few examples of distributions and quantiles that satisfy these assumptions."
>
> Concerning our technical assumptions, we provide below a thorough discussion, which also appears in the rebuttal for Reviewer 49vS. This is an important point, and we plan to add it to the final version.
>
> We believe we can safely characterize the assumptions from Definition 2 as minimal for the following reasons. First, assumptions (1) and (3) are indeed necessary for private quantile estimation. This follows from the tight lower bound for the one-dimensional case (see the main result of [TVGZ20]), where it is established that the minimal sample complexity to privately learn one quantile of a distribution explodes to infinity whenever there is no bound on the possible quantile values ($R=R_{\max}=\infty$) or the distribution has no mass around the quantile ($L=0$). We highlight that the latter (having a positive density around the quantile) is a requirement for optimal estimation rates (and a standard assumption) even for non-private 1D quantile estimation (e.g., see reference [1, Chs. 21, 25.3] in the Rebuttal for Reviewer 49vS). Now, assumption (2) is equivalent to the assumption that the floating body contains a ball of radius $R_{\min}$. Again, if $R_{\min}=0$, the floating body has an empty interior, and the floating body's estimation becomes a degenerate task. Finally, assumption (4) is a very mild boundedness technical condition that our data is arbitrarily polynomially-bounded, which almost does not hurt generality. For example, for the "heavy-tailed" Cauchy random vectors we can take the polynomial $dn$.
>
> Regarding examples of distributions and quantiles that satisfy these assumptions. Section E in the supplementary material is devoted to such an example, symmetric log-concave measures (which includes the Gaussian and Laplace distributions, among many others). Another family of classes that satisfy our assumptions and for which we can derive meaningful bounds is that of $\alpha$-stable distributions (see [Definition 5, AR20]). We can add the details to the final version if you think this additional example is needed.

---

### Official Review · Reviewer_zzac · 2022-07-10

**Rating:** 7
**Confidence:** 3
**Soundness:** 3 good
**Presentation:** 3 good
**Contribution:** 3 good

**Summary:**

This paper looks at the problem of privately estimating the properties of the convex float body. Three different properties are considered: 1. privately estimating many-quantiles estimation; 2. private interior point; 3. private uniform sampling from the floating body. Furthermore, a nice framework of estimating approximate holder queries is proposed, which should be of own interest.

**Questions:**

Is Proposition 1 technically correct? Do not we need some condition on Lipschitz?

**Strengths And Weaknesses:**

Strengths:
1. This paper looks at the problem of estimating properties of the convex body. Several upper bounds have been provided, which strictly improve the previous result.
2. A framework of estimating approximate holder queries is proposed, which may have other applications.
3. I like the idea of designing a Lipschitz estimator on a "typical dataset" and then extending it to all the possibilities of datasets by Lipschitz extension.

Weakness:
1. No lower bounds have been provided, making it hard to evaluate how "well" the estimators are.
2. Many mathematical definitions are used in this work, making it a bit hard to follow. I understood that it was unavoidable in proposing a general framework. But adding more languages and intuitions will be more friendly to the readers.

---

> ### Author Response · Authors · 2022-08-02
> **Rebuttal for Reviewer zzac**
>
> Thank you for your positive feedback and helpful comments. The Extension Lemma is indeed a powerful tool in the design of differentially private algorithms, and we hope it may find further uses.
>
> Below we address your comments and questions.
>
> - "No lower bounds have been provided, making it hard to evaluate how "well" the estimators are."
>
> Indeed, our work focuses on proving new upper bounds; specifically, proving that various private high dimensional statistical tasks are possible with polynomially many samples.
>
> Establishing tight bounds for the rate of private statistical estimators in high dimensions is one of the most important and central challenges in the current privacy literature, and seems difficult in many settings at this point. This is especially true when we go beyond the two more well-studied regimes (the "strong assumptions" regime, such as assuming (sub-)Gaussianity, and the "worst-case regime"). Notably, even in the worst-case regime, for problems such as private interior point [BMNS19] or privately learning halfspaces [KMST20] that are close to those we study in this paper there are still significant gaps between the known upper and lower bounds (see, e.g., Section 6 in [KMST20]).
>
> For the "mild assumptions" setting considered in this work, the only applicable lower bound we are aware of applies in the restrictive $d=1$-dimensional case where the goal is to estimate a single ($M=1$) quantile, see [TVGZ20]. Generalizing their lower bound arguments is a highly non-trivial mathematical task, which requires answering some open challenging questions on the geometry of quantiles in high-dimensions. We consider establishing tight lower bounds and potentially answering these geometric questions a fascinating and challenging direction for future work.
>
>
>
> - "Many mathematical definitions are used in this work..."
>
> We agree that some parts of the paper are mathematically dense. Indeed, this is unavoidable due to the depth of the problem we address and the complexity and generality of our methods. While we have tried to make the paper as accessible as possible, in the final version, we plan to expand on the intuition behind our definitions. We hope that such additions may improve readability. We are also happy for any further suggestions from the reviewers on how to improve the presentation.
>
> - "Is Proposition 1 technically correct? Do not we need some condition on Lipschitz?"
>
> Yes, it is in fact correct at this level of generality. Note that the proposition assumes the existence of an $\varepsilon$-DP algorithm on a restricted set, and this is the sole assumption needed for the extension of the algorithm to exist. In this setting, $\varepsilon$-DP is the Lipschitz condition and the algorithm therefore can be perceived as a Lipschitz map: from samples (equipped with the Hamming distance in our case) to distributions (equipped with the infinity Renyi divergence). This is the reason Proposition 1 is called the (Lipschitz) extension lemma. We elaborate on this point in Section F in the supplementary material.

---

> > ### Comment · Reviewer_zzac · 2022-08-08
> > **Thank you for the response**
> >
> > Thank you for the response, which has resolved my questions.

---

### Official Review · Reviewer_Ntuo · 2022-07-18

**Rating:** 6
**Confidence:** 2
**Soundness:** 3 good
**Presentation:** 4 excellent
**Contribution:** 3 good

**Summary:**

This paper studies sample complexity for estimating quantiles in high dimensions, which is called the convex floating body (Turkey depth at least 1-q). It shows polynomial sample complexity of a differentially private algorithm for querying generic Holder statistics of the floating body with respect to the \delta_q norm. The result is under mild distributional assumptions. The general approach relies on the appropriate use of Extension Lemma to ensure privacy on all possible inputs rather than only the typical inputs. Following the general results, it gives three applications: private many-quantiles estimation, private interior point, and private sampling from the floating body.

**Questions:**

For the three applications, the theorems statement are like ``There exists an \epsilon-DP algorithm ...''. Can you elaborate on how to design the algorithm to achieve these bounds?

**Limitations:**

Yes.

**Strengths And Weaknesses:**

Strengths: The studied problem is interesting and important. It's a well-written paper and I enjoyed reading it. I appreciate the general results along with the three examples to make it more accessible. I'm not very familiar with private high-dimensional estimation literature. The paper provides a detailed discussion with relevant papers so that it makes it easier for me to understand the contribution. Moreover, I think the technical contribution is strong. I didn't read the appendix but the proof sketch is sound to me. In all three examples, the results improve upon the best-known previous results.

Weaknesses: I have one question listed below.

---

> ### Author Response · Authors · 2022-08-02
> **Rebuttal for Reviewer Ntuo**
>
> Thank you for the feedback. We have tried our best to make the paper as accessible as possible. Given the technical overhead required by our techniques, this was a non-trivial task, and we are very happy that you found the paper technically strong and enjoyable to read.
>
> Regarding the private algorithm achieving the main results of our paper: we highlight the main ideas of our generic algorithm in Section 1.3. The generic algorithm is applicable in a relatively wide setting (see Theorem 7) and its instantiation to each of the three main applications revolves around plugging in the right problem-specific estimator for each application (such as the Steiner point for Theorem 4).
>
> A much more detailed explanation of the algorithm appears in Sections B.3 and F of the supplementary material, and we hope that the details in these sections will help answer your question. In brief, the idea is to identify a 'typical' set where we can exploit distributional properties of the input in order to define a private algorithm, restricted to this set. Within the typical set, the private (and therefore randomized) algorithm is defined explicitly via its density (Equation (19) in Section B.3). To define the algorithm on the entire space, we use a Lipschitz extension type result (Proposition 1), which requires to solve a non-trivial optimization problem (Equation (27) in Section F).
>
> Please let us know if there is any detail you think we should emphasize or explain in Section 1.3, or if you would like us to elaborate further on any particular topic.

---

> > ### Comment · Reviewer_Ntuo · 2022-08-09
> > **response**
> >
> > Thank you for the response, which resolves my question.

---

### Author Response · Authors · 2022-08-02
**General Response**

Thank you all for your positive and helpful reviews. We are happy the reviewers found our technical contribution significant and our writing clear (especially given the depth and complexity of tools from convex geometry). Your comments and advice will be very helpful in making the exposition and contributions as accessible as possible.

We addressed separately the comments and questions of each reviewer.
One question asked by two reviewers concerns the nature of our assumptions, their minimality, and examples of distributions satisfying these assumptions. We provided the response in the rebuttals dedicated to each of these two reviewers.

---

### Meta-Review · Area_Chair_24Ky · 2022-08-29

**Recommendation:** Accept
**Confidence:** Certain

**Metareview:**

Estimating quantiles of a dataset is a fundamental problem. This is a challenging and interesting question in high dimensions. The authors use convex geometry to obtain algorithms for private estimation of quantiles. The reviewers all liked the work and the presentation.

A comment from me: In the second paragraph of the abstract, you use "relies upon deep robustness results". The word deep here is highly subjective, and while one can use it in a talk, it does not fit nicely in the abstract.

**Award:**

No

---

### Decision · Program_Chairs · 2022-09-14

Accept